# A dysmorphic mouse model reveals developmental interactions of chondrocranium and dermatocranium

**Susan M Motch Perrine[1]\*[†], M Kathleen Pitirri[1†], Emily L Durham[1], Mizuho Kawasaki[1], Hao Zheng[2], Danny Z Chen[2], Kazuhiko Kawasaki[1‡], Joan T Richtsmeier[1]\*[‡]**

[1]Department of Anthropology, The Pennsylvania State University, University Park, United States; [2]Department of Computer Science and Engineering, University of Notre Dame, Notre Dame, United States

**\*For correspondence:**
qzk2@psu.edu (SMMP);
jta10@psu.edu (JTR)

[†]These authors contributed equally to this work
[‡]These authors also contributed equally to this work

**Competing interest:** The authors declare that no competing interests exist.

**Abstract** The cranial endo and dermal skeletons, which comprise the vertebrate skull, evolved independently over 470 million years ago and form separately during embryogenesis. In mammals, much of the cartilaginous chondrocranium is transient, undergoing endochondral ossification or disappearing, so its role in skull morphogenesis is not well studied and it remains an enigmatic structure. We provide complete 3D reconstructions of the laboratory mouse chondrocranium from embryonic day (E) 13.5 through E17.5 using a novel methodology of uncertainty-guided segmentation of phosphotungstic enhanced 3D micro-computed tomography images with sparse annotation. We evaluate the embryonic mouse chondrocranium and dermatocranium in 3D, and delineate the effects of a *Fgfr2* variant on embryonic chondrocranial cartilages and on their association with forming dermal bones using the *Fgfr2c*$^{C342Y/+}$ Crouzon syndrome mouse. We show that the dermatocranium develops outside of and in shapes that conform to the chondrocranium. Results reveal direct effects of the *Fgfr2* variant on embryonic cartilage, on chondrocranium morphology, and on the association between chondrocranium and dermatocranium development. Histologically, we observe a trend of relatively more chondrocytes, larger chondrocytes, and/or more matrix in the *Fgfr2c*$^{C342Y/+}$ embryos at all timepoints before the chondrocranium begins to disintegrate at E16.5. The chondrocrania and forming dermatocrania of *Fgfr2c*$^{C342Y/+}$ embryos are relatively large, but a contrasting trend begins at E16.5 and continues into early postnatal (P0 and P2) timepoints, with the skulls of older *Fgfr2c*$^{C342Y/+}$ mice reduced in most dimensions compared to *Fgfr2c*$^{+/+}$ littermates. Our findings have implications for the study and treatment of human craniofacial disease, for understanding the impact of chondrocranial morphology on skull growth, and potentially on the evolution of skull morphology.

## Editor's evaluation

Richtsmeier and colleagues demonstrate that chondrocranium and dermatocranium development are associated and that mutations in Fgfr significantly alter skull shape in part via the chondrocranium by means of a 3D modeling technique. The study is inspiring in providing new data regarding the role of the chondrocranium in normal craniofacial development and shedding light on the putative correspondence between chondrocranial elements and dermal skull bones. This work will be of interest to readers in the fields of vertebrate developmental biology, evolutionary anatomy, genetic disease, and vertebrate paleontology.

## Introduction

The heads of modern vertebrates arose as a protective, predominantly cartilaginous assembly that surrounded the major cranial organs of early vertebrates. The emergence of the cranial endoskeleton was followed by the appearance of the cranial dermal skeleton 470 Mya or earlier (*Janvier, 2015*; *Janvier, 1993*; *Sansom and Andreev, 2019*). The cranial endoskeleton includes the cartilaginous chondrocranium and pharyngeal skeleton that form prior to adjacent cranial dermal bones of the dermatocranium (*de Beer, 1937*; *Kawasaki and Richtsmeier, 2017a*; *Kawasaki and Richtsmeier, 2017b*; *Pitirri et al., 2020*). Though elements of these two skeletal systems have changed drastically over evolutionary time (*Janvier, 1993*; *Schultze, 1993*; *Zhu et al., 2013*), their association has been maintained, excepting in Chondrichthyes who secondarily lost their dermal skeleton (*Schultze, 1993*). Most modern vertebrate skulls are composite structures formed by the union of the endo and dermal (exo) cranial skeletons that form embryonically and/or evolutionarily in cartilage and bone, respectively, and evolved as distinct systems (*Hirasawa and Kuratani, 2015*; *Jarvik, 1980*; *Patterson, 1977*). Based on our characterization of the mouse chondrocranium as a scaffold for cranial dermal bones (*Kawasaki and Richtsmeier, 2017a*), we test the hypothesis that prenatal development of the chondrocranium and dermatocranium of modern mammals is integrated by analyzing this relationship in a mouse model for a human craniofacial disease. We propose that chondrocranial morphology directly impacts the formation of cranial dermal bones until cartilages dissolve or are mineralized endochondrally.

Elements of the mouse chondrocranium form individually in sequence beginning at embryonic day 12.5 (E12.5), fuse to provide an intricate protective covering for the brain and other sense organs, and parts of these elements begin to dissolve by E16.5 (*Pitirri et al., 2020*). Though many chondrocranial elements are transient, no modern vertebrate species has lost the chondrocranium during evolution suggesting its essential role in skull development (*Kawasaki and Richtsmeier, 2017a*). Observed variation in chondrocranial anatomy across species (*de Beer, 1937*) indicates its contribution to phylogenetic differences in skull morphology. Dermal bones of the skull arise individually in association with chondrocranial cartilages (*Kawasaki and Richtsmeier, 2017a*; *Pitirri et al., 2020*) but are ultimately joined with other bones by sutures that serve as essential sites of bone formation and growth (*Opperman, 2000*). During growth, mesenchyme of the suture keeps adjacent bones separated while osteoprogenitor mesenchymal cells within the osteogenic fronts of these bones proliferate and differentiate into osteoblasts that mineralize osteoid by intramembranous ossification (*Farmer et al., 2021*; *Holmes et al., 2021*; *Iseki et al., 1997*; *Opperman, 2000*). In craniosynostosis, a condition that always involves premature fusion of cranial suture(s) and can include additional postcranial and craniofacial anomalies, osteoblasts mineralize the suture before the completion of brain growth, alter subsequent growth patterns of cranial dermal bone, and produce abnormal head shapes (*Flaherty et al., 2016*).

Approximately 1 in 2000–2500 children of all ethnic groups are born with craniosynostosis conditions (*Heuzé et al., 2014*; *Lajeunie et al., 2006*) and though variants of many genes are associated with these disorders (*Cuellar et al., 2020*; *Calpena et al., 2020*; *Goos and Mathijssen, 2019*; *Holmes et al., 2021*; *Justice et al., 2012*; *Maruyama et al., 2021*; *Wilkie, 1997*; *Wilkie and Morriss-Kay, 2001*), alteration to the function of fibroblast growth factor receptor 2 (FGFR2) results in the more common craniosynostosis syndromes of Apert, Crouzon, and Pfeiffer. Though nearly all individuals affected with each of these syndromes have premature suture closure, the distinctive set of non-sutural phenotypes that comprise each syndrome depicts craniosynostosis as a complex growth disorder affecting multiple cranial tissues whose development is targeted by variants in ways that remain poorly understood (*Flaherty et al., 2016*).

Because humans share key developmental mechanisms with most other mammals, mouse models for the more common craniosynostosis syndromes have provided an experimental system for the study of aberrant genetic signaling in embryonic craniofacial development. The $Fgfr2c^{C342Y/+}$ Crouzon syndrome mouse model (*Eswarakumar et al., 2004*) carries a cysteine to tyrosine substitution at amino acid 342 (Cys342Tyr; C342Y) in the protein encoded by $Fgfr2c$ equivalent to the FGFR2 variant common to Pfeiffer and Crouzon syndromes (*Eswarakumar et al., 2004*; *Oldridge et al., 1995*; *Reardon et al., 1994*; *Rutland et al., 1995*). The FGFR2c C342Y variant is associated with constitutive activation of the receptor that increases osteoblast proliferation (*Eswarakumar et al., 2004*), may affect osteoblast differentiation at different stages of development (*Liu et al., 2013*; *Miraoui et al., 2009*), and is associated with craniofacial dysmorphogenesis and premature fusion of the coronal

suture, typically prenatally. In mice, *Fgfr2c* is required for normal function of osteoblast lineage cells and interacts with *Fgfr3,* important to cells in the chondrocyte lineage during endochondral osteogenesis (*Eswarakumar et al., 2004*; *Eswarakumar et al., 2002*).

The established explanation for cranial dysmorphogenesis in craniosynostosis syndromes is that premature closure of sutures results in changes in growth trajectories local to sutures of the growing skull (*Johnson and Wilkie, 2011*). Suture closure is considered the primary insult, initiating changes in growth patterns, and increased intracranial pressure that can harm the brain and other cranial organs. Under this hypothesis, research into mechanism underlying craniosynostosis has focused primarily on how changes in genetic regulation affect osteoblast function, dermal bone formation, and mineralization of cranial suture mesenchyme, while typical therapies involve corrective and/or reconstructive surgery to adjust the size, shape, and position of skull bones to improve appearance and function. The recent definition of sutures as a mesenchymal stem cell niche (*Maruyama et al., 2016*; *Park et al., 2016*; *Zhao et al., 2015*) provides a potential alternative approach to correcting closed sutures by combining biodegradable materials with mesenchymal stem cells to regenerate functional cranial sutures (*Yu et al., 2021*). However, skulls of mice carrying specific *Fgfr2* variants are dysmorphic prior to suture closure (*Motch Perrine et al., 2014*), cranial tissues other than bone are dysmorphic in these mice at birth (*Holmes et al., 2018*; *Martínez-Abadías et al., 2013*; *Motch Perrine et al., 2017*; *Peskett et al., 2017*), and a diversity of cell types are identified within the embryonic murine coronal suture by single cell transcriptome analysis (*Farmer et al., 2021*; *Holmes et al., 2021*). Investigation of the effect of aberrant FGF/FGFR signaling on the function of a variety of cranial cells and tissues is required to fully understand the pathogenesis of craniosynostosis conditions. The unique capacity of cartilage to grow interstitially enabling rapid, continuous growth in size and change in shape ensures customized protection for embryonic cranial organs prior to bone formation, and the established association between cranial cartilage and endochondral bone confirms the importance of chondrocranial elements to skull shape. Though not as extensively studied, the demonstrated temporospatial association between specific cranial cartilages and individual dermal bones during embryogenesis (*Kawasaki and Richtsmeier, 2017a*; *Pitirri et al., 2020*) suggests the potential for the chondrocranium to influence the position, size, shape, and development of dermal bones.

Our goal is to elucidate the developmental relationship between the chondrocranium and dermatocranium in *Fgfr2c*$^{C342Y/+}$ mice whose skull phenotype parallels that of humans with Crouzon/Pfeiffer syndrome with known deviation in FGF/FGFR signaling (*Eswarakumar et al., 2004*; *Martínez-Abadías et al., 2013*; *Perlyn et al., 2006*; *Snyder-Warwick et al., 2010*). The impact of this research is twofold: (1) the samples and novel methods for embryonic cartilage visualization (*Lesciotto et al., 2020*), and deep learning based segmentation using uncertainty-guided self-training with very sparse annotation (*Zheng et al., 2020*) allow us to address questions inaccessible in the study of humans but inform us about human craniofacial development and disease process; and (2) our 3D morphological analyses provide a unique opportunity for innovative evaluation of the effect of a variant on embryonic cranial cartilage formation and on the relationship between chondrocranial cartilage and dermal bone formation. Since it is known that the prenatal dermatocranium is dysmorphic in these mice, three outcomes are possible: (i) chondrocranial morphology of *Fgfr2c*$^{C342Y/+}$ mice and their controls (*Fgfr2c*$^{+/+}$ littermates) is similar indicating that the variant affects the cranial osteoblast lineage but not the chondrocyte series; (ii) chondrocranial morphology separates *Fgfr2c*$^{C342Y/+}$ and *Fgfr2c*$^{+/+}$ littermates but there is a lack of correspondence in the morphological effects on the dermatocranium and the chondrocranium indicating that the variant affects the chondrocyte series and the osteoblast lineage but that the two cranial skeletons are dissociated; or (iii) chondrocranial morphology differs between genotypes and the morphological effects of the variant on chondrocranial cartilages and dermatocranial bone show correspondence, indicating integration of chondrocranial and dermatocranial development. Our quantitative analyses demonstrate that the *Fgfr2c C342Y* mutation induces changes in chondrocranial cartilages that in turn, affect the development of cranial dermal bone. These results provide insight into the role of the chondrocranium in dermatocranium development in craniosynostosis and by extension, in normal development.

## Results

### Segmentation and visualization of embryonic mouse cranial bone and cartilage in 3D

Embryonic bone was segmented from 3D micro-computed tomography (microCT) images by thresholding techniques using Avizo 2020.2 (ThermoFisher Scientific, Waltham, MA), but segmenting embryonic cranial cartilage using deep learning based fully convolutional networks (FCNs) (*Long et al., 2015*; *Ronneberger et al., 2015*; *Zheng et al., 2019*) remains a challenging task. The difficulty involves a combined cadre of conditions including significant topological variation across cranial cartilages, large-size image volumes ($\bar{X} \approx 1300 \times 1700 \times 2000$ voxels), extremely thin regions-of-interest (ROIs), and unobtainability of voxel-wise annotation of whole volumes for network training. Our goal was to enable automated segmentation over developmental time, but full annotation (i.e. labeling all ROIs in a sufficient number of whole 3D volumes) for training deep learning based FCN models for chondrocranium segmentation is impractical. The reasons include large image size necessary to capture biological complexity, substantial changes in corresponding anatomical regions across developmental time and genotypes, and the need for sample sizes adequate to achieve statistical power. Consequently, a new two-phase approach implementing sparse annotation was used for training our segmentation model. The two-phase approach involves automatic segmentation of the chondrocranium with very sparse annotation to bridge the performance gap relative to full annotation and integration of limited human corrections to fine-tune the model. Our two-phase approach (https://github.com/ndcse-medical/CartSeg_UGST; *Sapkota, 2022*) is built on an automatic segmentation procedure (*Zheng et al., 2020*) that produced full 3D reconstructions of the chondrocranium from E13.5 through E17.5 for *Fgfr2c^{C342Y/+}* mice and their *Fgfr2c^{+/+}* littermates (*Figure 1*; *Figure 1—video 1*).

### The chondrocranium

#### Morphology of the mouse embryonic chondrocranium E13.5–E17.5

The appearance of the parachordal cartilages marks the initiation of the chondrocranium in C57BL/6 J mice at E12.5 (*Kawasaki and Richtsmeier, 2017a*; *Kawasaki and Richtsmeier, 2017b*) with the subsequent appearance and continual growth of additional chondrocranial cartilages (*Kawasaki and Richtsmeier, 2017a*; *Pitirri et al., 2020*). By E13.5, the lateral wall of the preoccipital region of *Fgfr2c^{C342Y/+}* mice consists of well-developed ala orbitalis (AO), sphenethmoid commissure (CSE), and tectum transversum (TTR), while *Fgfr2c^{+/+}* mice do not adequately develop these structures until E14.5 (*Figure 1*, *Figure 1—figure supplement 2*; see https://doi.org/10.25550/J-RHCA for interactive viewer of 3D reconstructions). The tectum nasi (TN), AO, and TTR are more developed and thicker in *Fgfr2c^{C342Y/+}* mice relative to *Fgfr2c^{+/+}* mice at E13.5, as shown by 3D thickness maps (*Figure 2A and D*; *Figure 2—figure supplement 1*) and cleared and stained specimens (*Figure 2B and C*; *Figure 2—figure supplement 1*). At E13.5, *Fgfr2c^{C342Y/+}* and *Fgfr2c^{+/+}* mice show a break in the brain case floor between the septum nasi (SN) and the hypophysis (*Figure 1C*; *Figure 1—figure supplement 1*; *Figure 2*; *Figure 2—figure supplement 1*). At E13.5, the AO and TTR extend further apically over the developing brain and are larger in *Fgfr2c^{C342Y/+}* mice relative to *Fgfr2c^{+/+}* mice, and the portion of the orbitoparietal commissure (COP) rostral to the TTR contains relatively more cartilage along its apical lip (*Figure 1*; *Figure 1—figure supplement 1*; Figure 2; *Figure 2—figure supplement 1*). This results in a broader and thicker rim of cartilage along the lateral wall, which in some *Fgfr2c^{C342Y/+}* individuals provides uninterrupted coverage of the lateral aspect of the preoccipital region (*Figure 1* and *Figure 2*). From E14.5 through E17.5, the AO and TTR appear thicker and extend more apically in *Fgfr2c^{C342Y/+}* relative to *Fgfr2c^{+/+}* mice, with more apical projections of thin parietal plate (PP) cartilage over time, even as skull bone mineralizes (*Figure 2B and C*; *Figure 2—figure supplements 2–5*). Most elements of the chondrocranium have formed by E15.5 (*Figure 1*, *Figure 1—figure supplement 3*; *Figure 2*; *Figure 2—figure supplement 3*). Endochondral ossification has not yet initiated at this age and dermatocranial elements are just beginning to form so the E15.5 skull is predominantly cartilaginous (*Figure 1—video 1*, *Figure 1—video 2*). Disintegration of portions of the chondrocranium is not evident at E15.5 but prior to E16.5, AO, TTR, and COP begin to dissolve in both genotypes, becoming thinner and taking on a lace-like appearance (*Figure 1C*; *Figure 1—figure supplement 3*; *Figure 1—figure supplement 4*; *Figure 2*; *Figure 2—figure supplement 3*; *Figure 2—figure supplement 4*). Though cartilage is disappearing in both genotypes at E17.5, cartilages of the *Fgfr2c^{C342Y/+}*

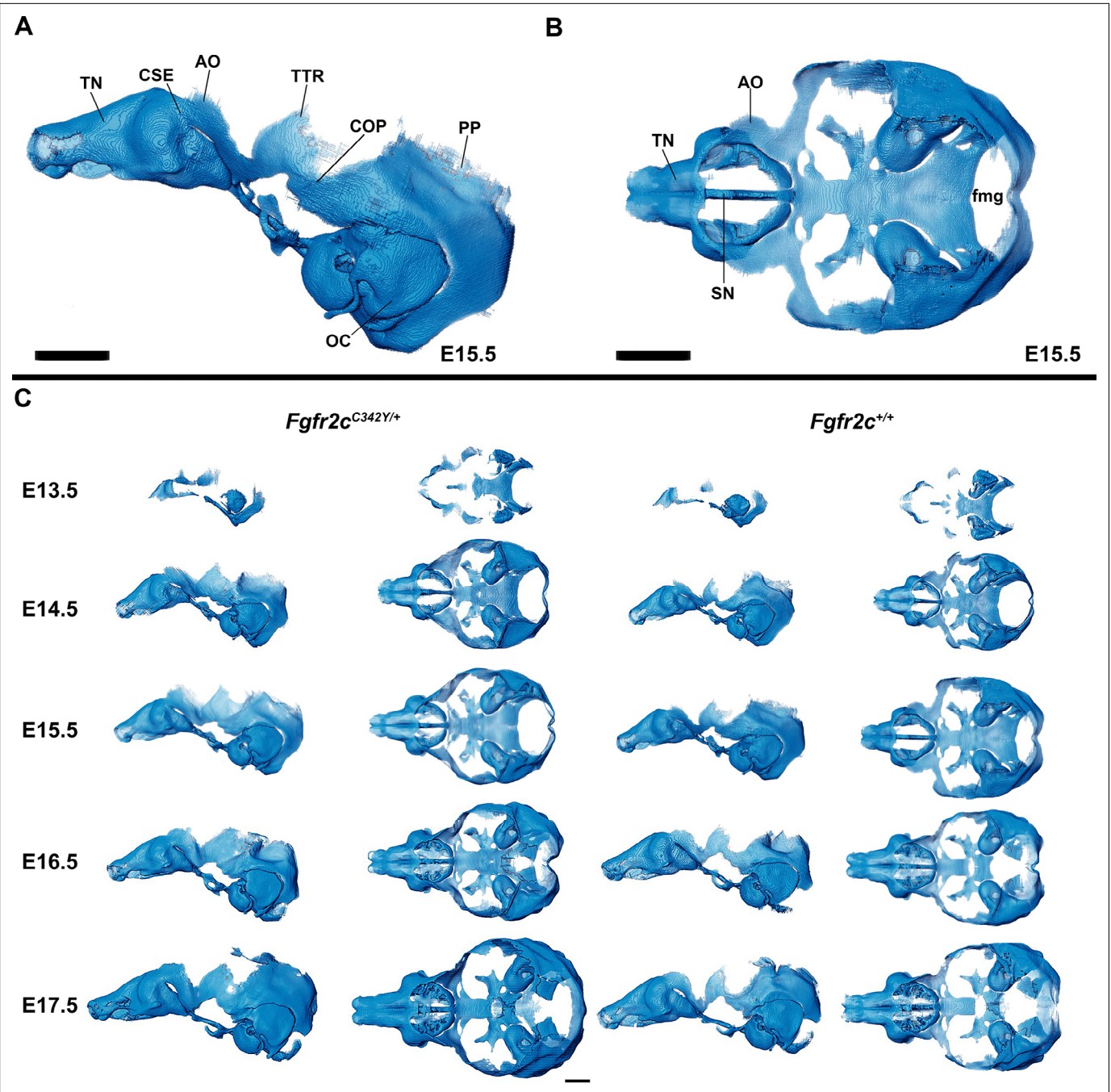

**Figure 1.** Anatomy of embryonic mouse chondrocranium. (**A, B**) At embryonic day 15.5 (E15.5), the *Fgfr2c^+/+* mouse chondrocranium, (**A**) lateral, and (**B**) superior views is complete, consisting of the olfactory region, braincase floor, and lateral walls of the preoccipital and occipital regions. Specific areas of interest include the ala orbitalis (AO), sphenethmoid commissure (CSE), otic capsule (OC), parietal plate (PP), septum nasi (SN), tectum nasi (TN), orbitoparietal commissure (COP), and tectum transversum (TTR) cartilages and the foramen magnum (fmg). (**C**) 3D reconstructions of *Fgfr2c^+/+* and *Fgfr2c^C342Y/+* chondrocrania from E13.5 to E17.5 in lateral and superior views with nasal capsule to the left. Scale bars = 1 mm. A cartoon of the mouse chondrocranium with more extensive anatomical labeling of cartilages and discussion of their development can be found in *Kawasaki and Richtsmeier, 2017a* and *Kawasaki and Richtsmeier, 2017b*. Interactive viewer of 3D reconstructions can be found at: https://doi.org/10.25550/J-RHCA.

The online version of this article includes the following video and figure supplement(s) for figure 1:

**Figure supplement 1.** Comparison of *Fgfr2c^C342Y/+*, (**A**) lateral and (**B**) superior views, and *Fgfr2c^+/+* (**C**) lateral and (**D**) superior view of mouse embryonic chondrocrania at embryonic day 13.5 (E13.5).

**Figure supplement 2.** Comparison of *Fgfr2c^C342Y/+*, (**A**) lateral and (**B**) superior views, and *Fgfr2c^+/+* (**C**) lateral and (**D**) superior view of mouse embryonic chondrocrania at embryonic day 14.5 (E14.5).

*Figure 1 continued on next page*

*Figure 1 continued*

**Figure supplement 3.** Comparison of *Fgfr2c^{C342Y/+}*, (**A**) lateral and (**B**) superior views, and *Fgfr2c^{+/+}* (**C**) lateral and (**D**) superior view of mouse embryonic chondrocrania at embryonic day 15.5 (E15.5).

**Figure supplement 4.** Comparison of *Fgfr2c^{C342Y/+}*, (**A**) lateral and (**B**) superior views, and *Fgfr2c^{+/+}* (**C**) lateral and (**D**) superior view of mouse embryonic chondrocrania at embryonic day 16.5 (E16.5).

**Figure supplement 5.** Comparison of *Fgfr2c^{C342Y/+}*, (**A**) lateral and (**B**) superior views, and *Fgfr2c^{+/+}* (**C**) lateral and (**D**) superior view of mouse embryonic chondrocrania at embryonic day 17.5 (E17.5).

**Figure 1—video 1.** Three-dimensional reconstruction of the isosurface of an embryonic day 15.5 (E15.5) *Fgfr2c^{+/+}* mouse chondrocranium.
https://elifesciences.org/articles/76653/figures#fig1video1

**Figure 1—video 2.** Three-dimensional reconstruction of the superimposed isosurfaces of an embryonic day 15.5 (E15.5) *Fgfr2c^{+/+}* mouse chondrocranium and skull.
https://elifesciences.org/articles/76653/figures#fig1video2

chondrocrania remain more complete relative to *Fgfr2c^{+/+}* mice (***Figure 1C***; ***Figure 1—figure supplement 5***; ***Figure 2***; ***Figure 2—figure supplement 5***). After E17.5, additional parts of the chondrocranium begin or continue to thin and disappear in both genotypes as the dermatocranium thickens and expands.

We used a suite of landmarks whose 3D coordinates (landmark coordinate data provided at DOI 10.26207/qgke-r185) could be reliably located across embryonic age groups (***Table 1***) to estimate differences in chondrocranial morphology. We analyzed three distinct configurations of 3D landmark coordinates representing cartilages of the nasal capsule, of the braincase floor, and of the lateral walls and roof of the vault using Euclidean Distance Matrix Analysis (EDMA) (***Lele and Richtsmeier, 2001***) (see Experimental Procedures section). Since the number of landmarks exceeds the sample size for these age groups, direct testing of the hypothesis of shape differences between chondrocrania of the two genotypes is not reported. Instead, confidence intervals ($\alpha=0.10$) for form difference estimators based on EDMA were implemented using the model independent bootstrap method (***Lele and Richtsmeier, 1995***). Confidence intervals were used to ascertain statistically significant estimates of localized morphological differences between genotypes with a statement regarding their accuracy.

At E13.5, delayed development of some structures made acquisition of all landmarks impossible and sample sizes were small (N=3), so confidence intervals are not reported. Still, 77% of all linear distances were larger in *Fgfr2c^{C342Y/+}* chondrocrania at E13.5, and of those, 40% showed increased size in *Fgfr2c^{C342Y/+}* mice ranging from 5 to 46%. By E14.5, over half of the linear distances among chondrocranial landmarks are 5–30% larger in *Fgfr2c^{C342Y/+}* mice. Local differences vary in magnitude at E14.5, and not all differences are statistically significant, but data indicate a sustained, global increase in size of *Fgfr2c^{C342Y/+}* chondrocrania relative to *Fgfr2c^{+/+}* littermates. By E15.5, measures that summarize the entire chondrocranium are relatively larger in *Fgfr2c^{C342Y/+}* mice as shown by confidence interval (***Figure 3A and C***; ***Figure 3—video 1***) and remain that way through E16.5. This difference becomes more localized with development so that by E17.5, significant differences are concentrated in the lateral walls of the preoccipital region extending to the posterior aspect of olfactory capsule (***Figure 3D and F***; ***Figure 3—video 2***).

For all ages considered, linear distances that measure the width and rostrocaudal length of the walls of the pre- and post-occipital regions are larger in *Fgfr2c^{C342Y/+}* mice relative to *Fgfr2c^{+/+}* littermates. The apical height of the TTR is relatively increased at all ages in *Fgfr2c^{C342Y/+}* mice (***Figure 3A and D***) and excess cartilage is deposited along the apical edge of the COP (***Figure 1C***; ***Figure 1—figure supplement 4***; ***Figure 2***; ***Figure 2—figure supplements 1–5***). Select cartilages of the braincase floor are statistically larger in *Fgfr2c^{C342Y/+}* mice at E14.5 (ranging from 4 to 7% larger) but the magnitude of differences of braincase floor dimensions between genotypes diminishes with age, with fewer statistically significant differences between genotypes at E15.5, E16.5, and E17.5. The olfactory capsule is significantly larger in nearly all dimensions in *Fgfr2c^{C342Y/+}* mice at E14.5, with some dimensions being as much as 25% larger relative to *Fgfr2c^{+/+}* littermates. The exception is the area described by the landmarks that delineate the superior surface of the posterior nasal capsule (landmarks: rncse, lncse, psep; landmark coordinate data available at DOI 10.26207/qgke-r185), which is consistently smaller in *Fgfr2c^{C342Y/+}* mice, though not statistically significantly smaller until E16.5. Excepting these dimensions, the olfactory capsule of *Fgfr2c^{C342Y/+}* mice remains relatively large through E17.5, though

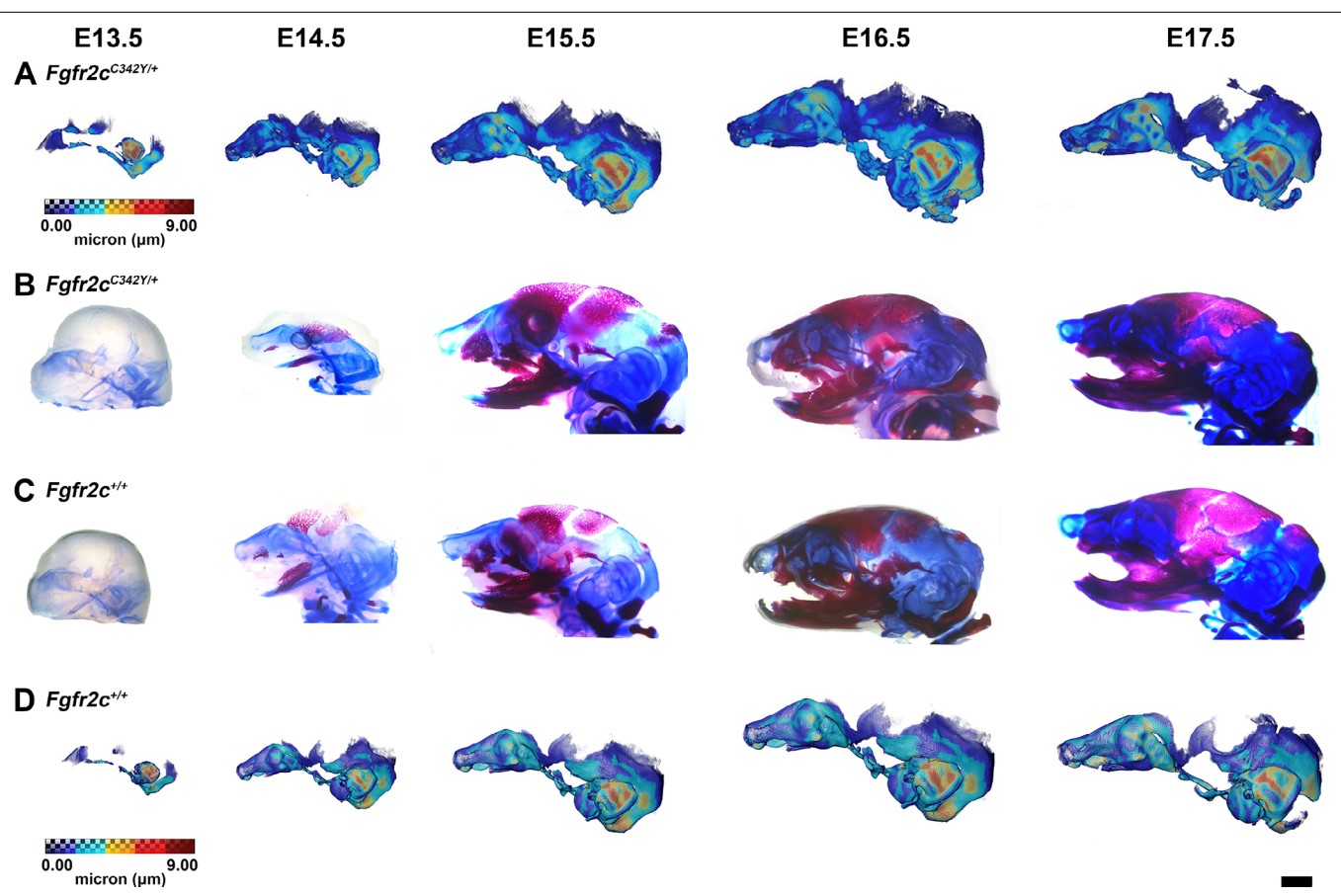

**Figure 2.** Thickness maps of the chondrocranium of mice segmented from PTA-enhanced micro-computed tomography (microCT) images and similarly aged, cleared, and stained specimens, embryonic day 13.5–17.5 (E13.5–E17.5). (**A, D**) Thickness maps of *Fgfr2c$^{C342Y/+}$* (**A**) and *Fgfr2c$^{+/+}$* (**D**) mice segmented from PTA-enhanced microCT images. Colormap indicates cartilage thickness that ranged from just over 0 μm (dark blue) to nearly 9 μm (dark red). (**B, C**) *Fgfr2c$^{C342Y/+}$* (**B**) and *Fgfr2c$^{+/+}$* (**C**) specimens that were chemically cleared are stained with Alcian blue indicating proteoglycans in cartilage and alizarin red indicating calcium deposits. Scale bar = 1 mm.

The online version of this article includes the following figure supplement(s) for figure 2:

**Figure supplement 1.** Left lateral view of thickness maps of the chodrocrania of mice segmented from phosphotungstic acid (PTA)-enhanced micro-computed tomography (microCT) images of *Fgfr2c$^{C342Y/+}$* (**A**) and *Fgfr2c$^{+/+}$* (**B**) mice and cleared and stained *Fgfr2c$^{C342Y/+}$* (**C**) and *Fgfr2c$^{+/+}$* (**D**) mice at embryonic day 13.5 (E13.5).

**Figure supplement 2.** Left lateral view of thickness maps of the chodrocrania of mice segmented from phosphotungstic acid (PTA)-enhanced micro-computed tomography (microCT) images of *Fgfr2c$^{C342Y/+}$* (**A**) and *Fgfr2c$^{+/+}$* (**B**) mice and cleared and stained *Fgfr2c$^{C342Y/+}$* (**C**) and *Fgfr2c$^{+/+}$* (**D**) mice at embryonic day 14.5 (E14.5).

**Figure supplement 3.** Left lateral view of thickness maps of the chodrocrania of mice segmented from phosphotungstic acid (PTA)-enhanced micro-computed tomography (microCT) images of *Fgfr2c$^{C342Y/+}$* (**A**) and *Fgfr2c$^{+/+}$* (**B**) mice and cleared and stained *Fgfr2c$^{C342Y/+}$* (**C**) and *Fgfr2c$^{+/+}$* (**D**) mice at embryonic day 15.5 (E15.5).

**Figure supplement 4.** Left lateral view of thickness maps of the chodrocrania of mice segmented from phosphotungstic acid (PTA)-enhanced micro-computed tomography (microCT) images of *Fgfr2c$^{C342Y/+}$* (**A**) and *Fgfr2c$^{+/+}$* (**B**) mice and cleared and stained *Fgfr2c$^{C342Y/+}$* (**C**) and *Fgfr2c$^{+/+}$* (**D**) mice at embryonic day 16.5 (E16.5).

**Figure supplement 5.** Left lateral view of thickness maps of the chodrocrania of mice segmented from phosphotungstic acid (PTA)-enhanced micro-computed tomography (microCT) images of *Fgfr2c$^{C342Y/+}$* (**A**) and *Fgfr2c$^{+/+}$* (**B**) mice and cleared and stained *Fgfr2c$^{C342Y/+}$* (**C**) and *Fgfr2c$^{+/+}$* (**D**) mice at embryonic day 17.5 (E17.5).

**Table 1.** Anatomical definitions of chondrocranial landmarks used in EDMA comparisons and morphological integration analyses. Landmark locations can be visualized on a 3D reconstruction of the embryonic mouse chondrocranium at https://getahead.la.psu.edu/landmarks/.

| Landmark abbreviation | Landmark definition | Olfactory capsule landmarks used in euclidean distance matrix analysis (EDMA) | Braincase floor landmarks used in EDMA | Lateral wall and roof of preoccipital and occipital region landmarks used in EDMA | Lateral wall and roof of preoccipital region landmarks used in Morpholog-ical Integration analysis |
|---|---|:---:|:---:|:---:|:---:|
| asep | Most anterior point of the septum nasi | x | | | |
| lao | Most superolateral point on the ala orbitalis, left side | | | x | |
| laottr | Most superior point of the intersection of the ala orbitalis and tectum transversum, left side | | | x | x |
| lapnc | Most anterior point of the paraseptal cartilage, left side | | | | |
| lcsp | Intersection of the sphenocochlear comissure and pars cochlearis, left side | | x | | |
| llpca | Most lateral point on the pars canalicularis, left side | | x | | |
| llat | Most lateral point on the left ala temporalis, left side | | x | | |
| lncse | Most superior anterior point where the nasal capsule (pars intermedia) intersects with the sphenethmoid commissure, left side | x | | | x |
| lppi | Most lateral point on the prominent pars intermedia, left side | x | | | |
| lppnc | Most posterior point of the paraseptal cartilage, left side | x | | | |
| ltpoa | Intersection of the tectum posterious and occiptal arch on the foramen magnum, left side | | | x | |
| lttr | Most superior point on tectum transversum, left side | | | x | x |
| nct | Most posterior midoint at which the left and right nasal capsule connects with the trabecular cartilage | x | x | | |
| psep | Most posterior point of the septum nasi | x | | | x |
| rao | Most superolateral point on the ala orbitalis, right side | | | x | |
| raottr | Most superior point of the intersection of the ala orbitalis and tectum transversum, right side | | | x | x |
| rapnc | Most anterior point of the paraseptal cartilage, right side | x | | | |
| rcsp | Intersection of the sphenocochlear comissure and pars cochlearis, right side | | x | | |
| rlpca | Most lateral point on the pars canalicularis, right side | | x | | |
| rlat | Most lateral point on the ala temporalis, right side | | x | | |
| rncse | Most superior anterior point where the nasal capsule (pars intermedia) intersects with the sphenethmoid commissure, right side | x | | | x |
| rppi | Most lateral point on the prominent pars intermedia, right side | x | | | |
| rppnc | Most posterior point of the paranasal cartilage, right side | x | | | |
| rtpoa | Intersection of the tectum posterious and occiptal arch on the foramen magnum, right side | | | x | |
| rttr | Most superior point on tectum transversum, right side | | | x | x |

the magnitude of significant differences reduces with age, ranging from 5 to 15% (*Figure 3D and F*; *Figure 3—video 2*).

## Cellular characterization of embryonic cartilage of the chondrocranium

Observations of growth plate cartilages in long bones identify chondrocyte proliferation, hypertrophy, and matrix deposition as the cellular processes that contribute to cartilage growth (*Breur et al., 1991*; *Cooper et al., 2013*; *Wilsman et al., 2008*) while Kaucka and colleagues (*Kaucka et al., 2017*) proposed oriented clonal cell dynamics as the basis for accurate shaping of nasal cartilages. To

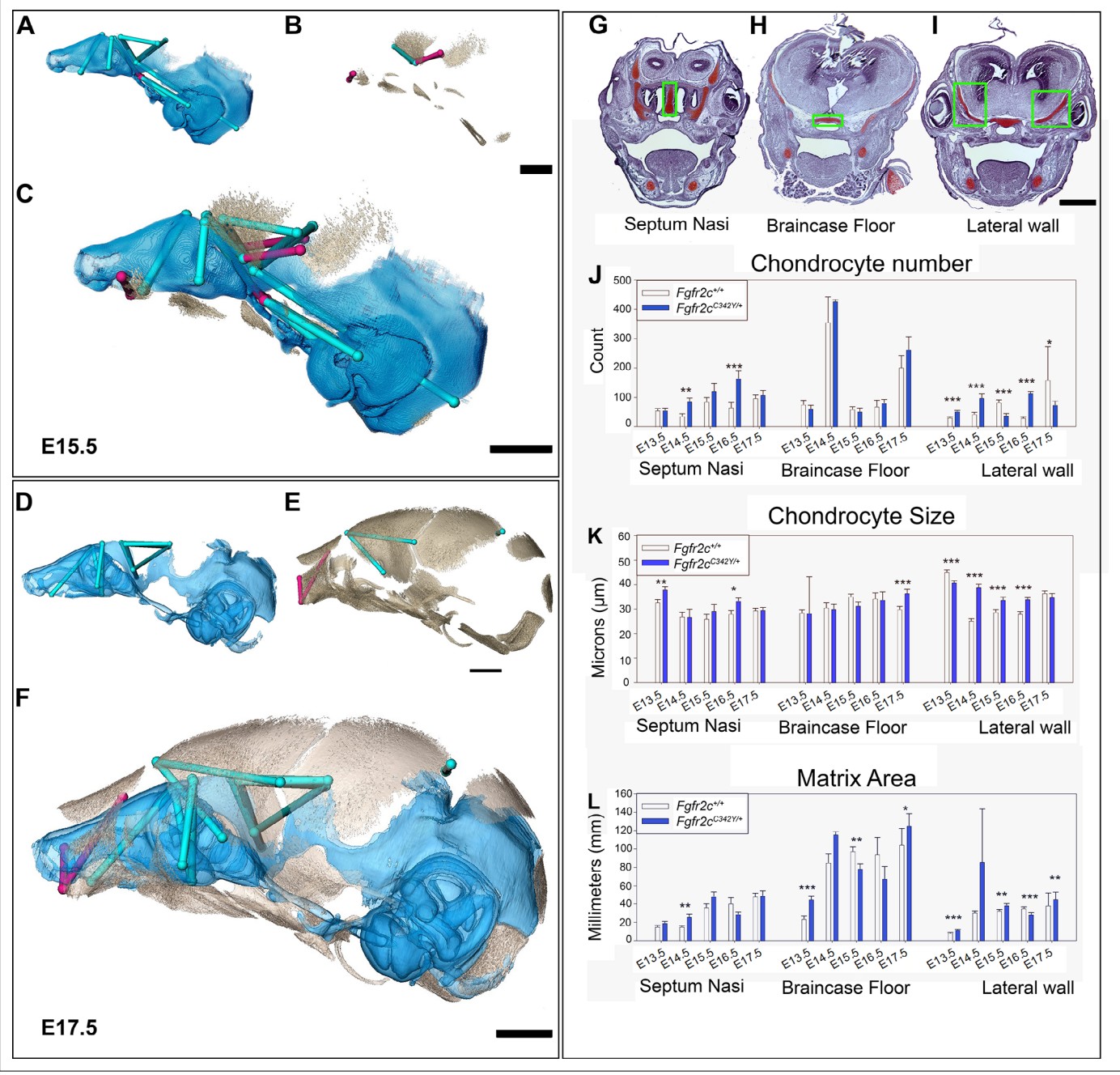

**Figure 3.** Euclidean distance matrix analysisDistance Matrix Analysis of the chondrocranium and bony skull, and histomorphology of the chondrocranium. Linear distances of the chondrocranium (**A, D**), bony skull (**B, E**), and the two superimposed (**C, F**) that are statistically significantly different between genotypes by confidence interval testing (α=0.10). Blue lines indicate linear distances that are significantly larger in *Fgfr2c$^{C342Y/+}$* mice; fuchsia lines are significantly reduced in *Fgfr2c$^{C342Y/+}$* mice. (**A–F**) Significant differences between chondrocranium and bony skulls of *Fgfr2c$^{C342Y/+}$* and *Fgfr2c$^{+/+}$*mice. A limited landmark set common to the chondrocranium and bony skull of embryonic day 15.5 (E15.5) (**A–C**) and E17.5 (**D–F**) embryos was used for analyses and indicated that the lateral wall and olfactory regions are most different between *Fgfr2c$^{C342Y/+}$* and *Fgfr2c$^{+/+}$* mice at these ages. (**G–L**) Histomorphology of the chondrocranium. Histological sections of the E15.5 chondrocranium highlighting the septum nasi (**G**), braincase floor (**H**), and lateral walls (**I**) in green boxes. These areas were assessed at E13.5, E14.5, E15.5, E16.5, and E17.5 for chondrocyte number (**J**), chondrocyte size (**K**), and area of cartilaginous matrix (**L**) in *Fgfr2c$^{C342Y/+}$* and *Fgfr2c$^{+/+}$* mice. In agreement with the larger chondrocrania of *Fgfr2c$^{C342Y/+}$* mice, there are localized regions that reveal increases in chondrocyte number, size, and/or contribution of matrix at each timepoint. Note the trend of increasing numbers of chondrocytes over time as expected in a growing chondrocranium. For histological analysis data are displayed as mean +/- standard error of at least three quantified images per individual (n) per region per age compared between genotypes using non-parametric Mann-Whitney U tests; *p≤0.05, **p≤0.01, ***p≤0.001. n (*Fgfr2c$^{+/+}$/Fgfr2c$^{C342Y/+}$*) = 4/4(E13.5), 7/7 (E14.5), 6/6 (E15.5), 6/5 (E16.5), 4/5 (E17.5). Scalebars = 1mm.

*Figure 3 continued on next page*

*Figure 3 continued*

The online version of this article includes the following video for figure 3:

**Figure 3—video 1.** Three-dimensional reconstruction of the superimposed isosurfaces of an embryonic day 15.5 (E15.5) $Fgfr2c^{+/+}$ mouse chondrocranium and skull with blue lines depicting linear distances that are significantly larger in $Fgfr2c^{C342Y/+}$ mice as compared to $Fgfr2c^{+/+}$ mice; fuchsia lines are significantly reduced in $Fgfr2c^{C342Y/+}$ mice as compared to $Fgfr2c^{+/+}$ mice.

https://elifesciences.org/articles/76653/figures#fig3video1

**Figure 3—video 2.** Three-dimensional reconstruction of the superimposed isosurfaces of an embryonic day 17.5 (E17.5) $Fgfr2c^{+/+}$ mouse chondrocranium and skull with blue lines depicting linear distances that are significantly larger in $Fgfr2c^{C342Y/+}$ mice as compared to $Fgfr2c^{+/+}$ mice; fuchsia lines are significantly reduced in $Fgfr2c^{C342Y/+}$ mice as compared to $Fgfr2c^{+/+}$ mice.

https://elifesciences.org/articles/76653/figures#fig3video2

investigate the cellular basis of morphological differences in chondrocranial morphology we analyzed the number and size of chondrocytes and the amount of matrix per region of interest in the SN, braincase floor, and the lateral walls of the chondrocranium in $Fgfr2c^{C342Y/+}$ mice relative to $Fgfr2c^{+/+}$ littermates at E13.5, E14.5, E15.5, E16.5, and E17.5 (**Figure 3G–L**). These three areas represent chondrocranial elements that either remain as cartilage in the adult (SN), ossify endochondrally (brain case floor), or disappear (lateral wall). We found significantly more chondrocytes in $Fgfr2c^{C342Y/+}$ SN at E14.5 (p=0.006) and E16.5 (p≤0.001) relative to $Fgfr2c^{+/+}$ littermates (**Figure 3J**). Chondrocytes in the septum nasi were larger in $Fgfr2c^{C342Y/+}$ mice at E13.5 (p=0.004) and E16.5 (p=0.016) (**Figure 3K**). The amount of matrix within the septum nasi was increased at E14.5 (p=0.003) in $Fgfr2c^{C342Y/+}$ mice relative to $Fgfr2c^{+/+}$ littermates (**Figure 3L**).

Histological analysis of braincase floor cartilage that mineralizes endochondrally indicates no changes in chondrocyte number between genotypes at any of the ages investigated, in agreement with our observation of similarity of 3D morphology of the braincase floor cartilages. Chondrocyte size was increased in $Fgfr2c^{C342Y/+}$ mice relative to $Fgfr2c^{+/+}$ littermates only at E17.5 (p=0.001) (**Figure 3K**). The amount of matrix was relatively increased in the braincase floor cartilage of $Fgfr2c^{C342Y/+}$ mice at E13.5 (p≤0.001) and E17.5 (p=0.042) but between these ages, at E15.5, the amount of matrix is relatively decreased in $Fgfr2c^{C342Y/+}$ mice (p=0.013) (**Figure 3L**).

Differences in the cartilages of the lateral walls contribute strongly to morphological differences between genotypes, reflecting the relatively early formation of these cartilages and their subsequent disintegration starting at E16.5 associated with dermal bone mineralization (especially the frontal and parietal bones) (**Kawasaki and Richtsmeier, 2017a**). Relatively more chondrocytes were identified in lateral wall cartilages of $Fgfr2c^{C342Y/+}$ mice at E13.5 (p≤0.001), E14.5 (p≤0.001), and E16.5 (p≤0.001), but at E15.5 and E17.5 there are more cells in the lateral walls of $Fgfr2c^{+/+}$ individuals (p≤0.001 and p=0.036, respectively) (**Figure 3J**). Cell size is relatively greater in $Fgfr2c^{+/+}$ lateral wall cartilages during early (E13.5, p≤0.001) prenatal development. Later, chondrocytes are relatively larger in $Fgfr2c^{C342Y/+}$ mice (E14.5 p≤0.001, E15.5 p=0.001, and E16.5 p≤0.001), consistent with the identification of a larger chondrocranium in $Fgfr2c^{C342Y/+}$ mice for these ages (**Figure 3K**). Area of cartilage matrix is greater in $Fgfr2c^{C342Y/+}$ mice at E13.5 (p≤0.001), E15.5 (p=0.010) and E17.5 (p=0.009). The relative increase in chondrocytes in the lateral wall cartilages of $Fgfr2c^{+/+}$ individuals at E15.5 is followed by an increase in cartilage matrix area in $Fgfr2c^{+/+}$ individuals at E16.5 (p≤0.001) (**Figure 3L**). Consequently, the significantly larger chondrocytes in $Fgfr2c^{C342Y/+}$ mice at E16.5 account for the observed relative increase in size of the lateral wall cartilages.

In sum, we observed a general trend of more chondrocytes, larger chondrocytes, and/or more matrix in the $Fgfr2c^{C342Y/+}$ mice as compared to their $Fgfr2c^{+/+}$ littermates at all timepoints prior to the disintegration of the chondrocranium that initiates just prior to E16.5. Localized differences are apparent across the cartilages we chose for study demonstrating that this is a complex system with mutually interactive characters (chondrocyte number, chondrocyte size, and matrix area) that react to the $Fgfr2c$ C342Y mutation in a location specific (septum nasi, braincase floor, lateral wall) and temporally sensitive manner.

## The bony skull
### Coronal suture fusion and bone volume

Initial mineralization of cranial dermal bone is apparent by alizarin red staining at E14.5 (**Figure 2B and C**; **Figure 2—figure supplement 2**), but individual cranial bones are not easily detected by microCT

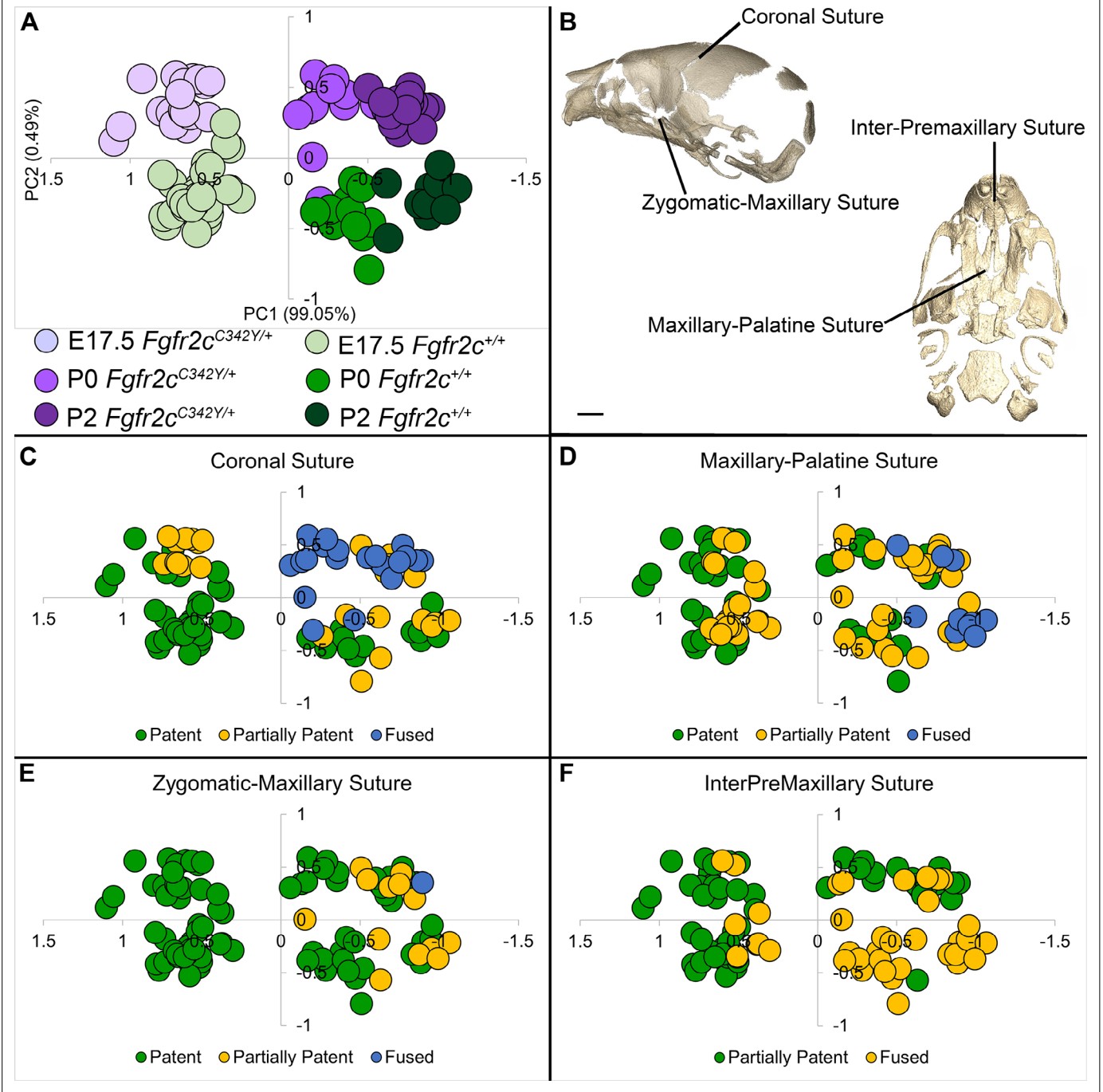

**Figure 4.** Relationship of suture patency patterns and craniofacial shape as estimated by principal components analysis (PCA). (**A**) PCA of skull linear distance data estimated from 3D landmark locations collected from micro-computed tomography (microCT) images of mice at E17.5, postnatal day 0 (P0), and P2 shows distribution of all individuals along principal component 1 (PC1) and PC2. (**B**) Suture patency was scored for sutures as visualized on left lateral and inferior views of a microCT 3D reconstruction of a $Fgfr2c^{+/+}$ P0 skull. (**C–F**) Distribution of individuals along PC1 and PC2 as shown in (**A**) coded for patency of the coronal suture (**C**), the maxillary-palatine suture (**D**), the zygomatic-maxillary suture (**E**), and the inter-premaxillary suture (**F**). Scale bar = 1 mm.

until E15.5 (*Figure 3B*). Using microCT, none of the mice show complete fusion of the coronal suture prior to birth (postnatal day 0 or P0) but half (9/18) of the $Fgfr2c^{C342Y/+}$ mice show bridging of one or both coronal sutures at E17.5, and by birth (P0), all $Fgfr2c^{C342Y/+}$ mice (11/11) show partial or complete closure of one or both coronal sutures (*Figure 4A-C*; bone microCT images, data used for PCA, and suture scores available at DOI 10.26207/qgke-r185). Coupled with evidence by alizarin red staining

**Table 2.** Bone volume summary statistics at embryonic day 17.5 (E17.5) and postnatal day 0 (P0) for *Fgfr2c*$^{C342Y/+}$ mice and their *Fgfr2c*$^{+/+}$ littermates.

Sample size (N) varied by availability of individual bone for analysis. Interfrontal and ethmoid bones develop relatively late and were not present in many specimens.

| Bone | E17.5 *Fgfr2c*$^{C342Y/+}$ N | Mean | S.D. | E17.5 *Fgfr2c*$^{+/+}$ N | Mean | S.D. | P0 *Fgfr2c*$^{C342Y/+}$ N | Mean | S.D. | P0 *Fgfr2c*$^{+/+}$ N | Mean | S.D. |
|---|---|---|---|---|---|---|---|---|---|---|---|---|
| Interparietal | 14 | 0.16 | 0.06 | 13 | 0.18 | 0.06 | 10 | 0.40 | 0.07 | 10 | 0.40 | 0.10 |
| Squamous occipital | 13 | 0.16 | 0.11 | 13 | 0.13 | 0.09 | 10 | 0.63 | 0.10 | 10 | 0.58 | 0.11 |
| Left lateral occipital | 14 | 0.43 | 0.06 | 13 | 0.42 | 0.05 | 10 | 0.65 | 0.06 | 10 | 0.59 | 0.07 |
| Right lateral occipital | 14 | 0.42 | 0.06 | 13 | 0.41 | 0.05 | 10 | 0.64 | 0.07 | 10 | 0.58 | 0.07 |
| Basioccipital | 14 | 0.58 | 0.08 | 13 | 0.53 | 0.06 | 10 | 0.88 | 0.09 | 10 | 0.74 | 0.10 |
| Left parietal | 14 | 0.34 | 0.08 | 13 | 0.36 | 0.09 | 10 | 0.71 | 0.09 | 10 | 0.68 | 0.14 |
| Right parietal | 13 | 0.35 | 0.09 | 13 | 0.37 | 0.08 | 10 | 0.73 | 0.09 | 10 | 0.69 | 0.15 |
| Left squamous temporal | 14 | 0.13 | 0.03 | 13 | 0.13 | 0.03 | 10 | 0.30 | 0.03 | 10 | 0.30 | 0.06 |
| Right squamous temporal | 14 | 0.12 | 0.03 | 13 | 0.12 | 0.03 | 10 | 0.32 | 0.04 | 10 | 0.30 | 0.06 |
| Left frontal | 14 | 0.74 | 0.16 | 13 | 0.66 | 0.12 | 10 | 1.30 | 0.13 | 10 | 1.14 | 0.19 |
| Right frontal | 14 | 0.74 | 0.16 | 13 | 0.65 | 0.12 | 10 | 1.28 | 0.13 | 10 | 1.13 | 0.18 |
| Interfrontal | 12 | 0.01 | 0.01 | 9 | 0.00 | 0.00 | 10 | 0.03 | 0.01 | 0 | 0 | 0 |
| Left maxilla | 14 | 0.48 | 0.10 | 13 | 0.48 | 0.08 | 10 | 0.93 | 0.14 | 10 | 0.82 | 0.15 |
| Right maxilla | 14 | 0.48 | 0.10 | 13 | 0.47 | 0.08 | 10 | 0.92 | 0.14 | 10 | 0.82 | 0.15 |
| Left jugal | 14 | 0.03 | 0.01 | 13 | 0.02 | 0.01 | 10 | 0.05 | 0.01 | 10 | 0.05 | 0.01 |
| Right jugal | 14 | 0.03 | 0.01 | 13 | 0.02 | 0.01 | 10 | 0.06 | 0.01 | 10 | 0.05 | 0.01 |
| Left nasal | 14 | 0.07 | 0.04 | 13 | 0.08 | 0.03 | 10 | 0.21 | 0.04 | 10 | 0.18 | 0.04 |
| Right nasal | 14 | 0.08 | 0.04 | 13 | 0.08 | 0.03 | 10 | 0.23 | 0.04 | 10 | 0.19 | 0.04 |
| Left premaxilla | 14 | 0.26 | 0.08 | 13 | 0.27 | 0.07 | 10 | 0.67 | 0.12 | 10 | 0.65 | 0.12 |
| Right premaxilla | 14 | 0.26 | 0.08 | 13 | 0.27 | 0.07 | 10 | 0.69 | 0.12 | 10 | 0.64 | 0.11 |
| Vomer | 14 | 0.09 | 0.02 | 13 | 0.07 | 0.01 | 10 | 0.16 | 0.04 | 10 | 0.13 | 0.03 |
| Left palatine | 14 | 0.23 | 0.05 | 13 | 0.20 | 0.03 | 10 | 0.42 | 0.07 | 10 | 0.36 | 0.06 |
| Right palatine | 14 | 0.23 | 0.05 | 13 | 0.20 | 0.04 | 10 | 0.42 | 0.06 | 10 | 0.36 | 0.05 |
| Presphenoid | 14 | 0.02 | 0.02 | 13 | 0.03 | 0.02 | 10 | 0.24 | 0.05 | 10 | 0.20 | 0.03 |
| Left sphenoid ala | 14 | 0.16 | 0.04 | 13 | 0.15 | 0.04 | 10 | 0.38 | 0.06 | 10 | 0.35 | 0.07 |
| Right sphenoid ala | 14 | 0.15 | 0.04 | 13 | 0.14 | 0.03 | 10 | 0.38 | 0.06 | 10 | 0.34 | 0.06 |
| Sphenoid body | 14 | 0.27 | 0.06 | 13 | 0.27 | 0.05 | 10 | 0.57 | 0.06 | 10 | 0.51 | 0.08 |
| Left petrous temporal | 14 | 0.03 | 0.01 | 13 | 0.03 | 0.01 | 10 | 0.25 | 0.10 | 10 | 0.31 | 0.11 |
| Right petrous temporal | 14 | 0.03 | 0.01 | 13 | 0.03 | 0.01 | 10 | 0.25 | 0.10 | 10 | 0.30 | 0.11 |
| Left mandible | 14 | 1.20 | 0.28 | 13 | 1.24 | 0.23 | 10 | 2.34 | 0.34 | 10 | 2.17 | 0.33 |
| Right mandible | 14 | 1.22 | 0.29 | 13 | 1.27 | 0.22 | 10 | 2.34 | 0.34 | 10 | 2.18 | 0.33 |
| Ethmoid | 0 | 0 | 0 | 0 | 0 | 0 | 8 | 0.02 | 0.02 | 10 | 0.03 | 0.02 |

of partially fused sutures at E17.5 by other investigators (**Peskett et al., 2017**) this confirms that coronal suture closure occurs between E17.5 and P0 in most *Fgfr2c*$^{C342Y/+}$ mice (**Martínez-Abadías et al., 2013**).

Bone size and volume are highly variable in both genotypes during prenatal development, but bone volume estimates reveal that some dermal bones (i.e. right nasal [p=0.043], left palatine [p=0.029], and right palatine [p=0.019]) and an endochondral bone (basioccipital, p=0.009) are significantly

larger in *Fgfr2c*^*C342Y/+* mice at P0 relative to *Fgfr2c*^*+/+* littermates. At E17.5, only the vomer (p=0.017) is significantly larger in *Fgfr2c*^*C342Y/+* mice relative to *Fgfr2c*^*+/+* littermates (*Table 2*).

## Morphometric comparison of pre- and post-natal *Fgfr2c*^*C342Y/+* Crouzon mouse bony skull

Skulls of adult *Fgfr2c*^*C342Y/+* mice show closure of the coronal sutures and small size (*Eswarakumar et al., 2004*), with a domed cranial vault and skull lengths reduced by as much as 20% (*Perlyn et al., 2006*). We used a suite of landmarks whose 3D coordinates (landmark coordinate data available at DOI 10.26207/qgke-r185) could be reliably located across embryonic age groups (*Table 3*) to explore differences in chondrocranial morphology from E17.5 to P2. Principal components analysis (PCA) of all linear distances among unique pairs of landmarks reveals that overall skull shape separates mice into groups consistent with developmental age and genotype (*Figure 4A*). Patency scoring of four cranial sutures was used to explore the relationship of suture closure patterns and morphological differences across developmental time (*Figure 4B–F*; PCA data and suture scores available at DOI 10.26207/qgke-r185).

We used EDMA (*Lele and Richtsmeier, 2001*) and three distinct configurations of 3D landmark coordinates representing bones of the facial skeleton, braincase floor, and lateral walls and roof of the cranial vault whose 3D coordinates could be reliably located across ages E15.5 through P2 (*Table 3*) to estimate differences in bony skull morphology (*Figure 3*; landmark data available at DOI 10.26207/qgke-r185). Confidence intervals (*α*=0.10) were implemented using the model independent bootstrap method to reveal statistically significant estimates of localized morphological differences between genotypes at E15.5, E16.5, E17.5, P0, and P2 along with a statement on their variability (*Lele and Richtsmeier, 1995*).

Though studies of adults have shown *Fgfr2c*^*C342Y/+* skulls to be significantly reduced in size, our analyses reveal that the bony skulls of *Fgfr2c*^*C342Y/+* embryos are generally larger than those of *Fgfr2c*^*+/+* littermates (*Martínez-Abadías et al., 2013*; *Motch Perrine et al., 2017*; *Table 4*; *Figure 3B, C, E and F*; *Figure 5A and D*). The lateral wall and roof of the cranial vault consist of dermal bones that show marked variability within and between genotypes at E15.5, likely due to differences in developmental timing among littermates (*Flaherty and Richtsmeier, 2018*). Dimensions of the *Fgfr2c*^*C342Y/+* frontal and parietal bones are significantly larger relative to *Fgfr2c*^*+/+* mice at E15.5, some by as much as 20%—but overall, the vault is nearly equal in length between genotypes. By E16.5 and continuing to E17.5, nearly all dimensions of the bones that make up the lateral walls and roof of the vault are larger in *Fgfr2c*^*C342Y/+* mice, indicating a pattern of relatively increased growth of these dermal bones in mice carrying the *Fgfr2c C342Y* mutation (*Figure 3B, C and E–F*; *Figure 3—video 1*; *Figure 3—video 2*). There are no significant differences in braincase floor morphology between genotypes at E15.5 but at E16.5 measures of bones of the braincase floor of *Fgfr2c*^*C342Y/+* mice become larger across all dimensions relative to *Fgfr2c*^*+/+* littermates. At E17.5, there are no significant differences between the two genotypes. Bones of the facial skeleton of both genotypes show marked variation at E15.5 resulting in few significant differences. Though not significantly different by confidence interval testing, dimensions of the developing maxilla are 5–15% larger in *Fgfr2c*^*C342Y/+* mice at E16.5. By E17.5, many dimensions of anterior dermal cranial vault bones remain larger in mice carrying the mutation, but the overall length of the *Fgfr2c*^*C342Y/+* vault is no longer larger anteroposteriorly relative to the vault of *Fgfr2c*^*+/+* mice, suggesting that bones of the posterior cranial vault are experiencing a distinct growth trajectory.

The increasing amount of mineralized bone with age enabled identification and use of a larger number of landmarks (K=24) for a comparative analysis of late embryonic (E17.5), newborn (P0), and early postnatal (P2) skull morphology between genotypes (*Figure 5*; *Table 3*). At E17.5, as the lateral walls of the chondrocranium dissolve but prior to coronal suture fusion, regional form difference (*Table 4*) and confidence interval testing (*Figure 5A and D*) reveal a generally larger facial skeleton surrounding the olfactory capsule, a shortened and narrowed anterior braincase floor, and an expanded posterior cranial base and vault in *Fgfr2c*^*C342Y/+* mice. This general pattern continues at P0 though the magnitude of the differences is reduced (*Figure 5B and E*). By P2, the height of the posterior cranial vault remains larger than normal (*Figure 5F*), as do measures of width of the lateral and occipital walls (*Figure 5C*), but all measures oriented along the rostrocaudal axis are relatively reduced in *Fgfr2c*^*C342Y/+* mice (*Figure 5C and F*). Select dimensions of the *Fgfr2c*^*C342Y/+* facial skeleton

**Table 3.** Anatomical definitions of bony skull (dermal bone and endochondral bone) landmarks used in Euclidean Distance Matrix Analysis (EDMA) and morphological integration analyses.

Landmark locations can be visualized on 3D reconstructions of the mouse skull at embryonic day 17.5 (E17.5) and postnatal day 0 (P0) https://getahead.la.psu.edu/landmarks/.

| | | Bony skull landmarks for ages E15.5, E17.5, P0, and P2 | | | | |
|---|---|---|---|---|---|---|
| Landmark description | | Anatomical region of interest | | | | |
| Landmark abbreviation | Landmark definition | Olfactory capsule landmark set used in EDMA of E15.5–P2 | Braincase floor landmark set used in EDMA of E15.5–P2 | Lateral wall and roof of pre-occipital and occipital region landmark set used in EDMA of E15.5–P2 | Lateral wall and roof of pre-occipital region landmark set used in Morphological Integration analysis | Global skull landmark set used in EDMA of E17.5, P0, and P2 |
| amsph | Most anterior-medial point on the body of the sphenoid | | | | | x |
| bas | Mid-point on the anterior margin of the foramen magnum, taken on basioccipital | | x | | | x |
| ethma | Anterior most point on the body of the vomer, taken on the ventral surface | | | | | x |
| intpar | Most anterior point on the ectocranial surface of the interparietal on the midsagittal plane | | | | | x |
| laif | Most anteroinferior point on the frontal bone, left side | | | x | x | |
| lalf | Most anteromedial point on the frontal bone, left side | | | | x | |
| lalp | Most anterolateral point on the palatine plate, left side | | | | | |
| lasph | Posteromedial point of the inferior portion of the left alisphenoid | | | | | x |
| lflac | Intersection of frontal process of maxilla with frontal and lacrimal bones, left side | | | | | x |
| lfppm | Most superoposterior point of the premaxilla accounting for the lateral part of the nasal aperature, left side | x | | | | x |
| liohd | Most distal point of the infraorbital hiatus, left side | x | | | | x |
| lnasapl | Most superoanterior point of the premaxilla accounting for the lateral part of the nasal aperture, left side | x | | | | x |
| loci | The superior posterior point on the ectocranial surface of occipital lateralis on the foramen magnum, left side | | x | x | | x |
| lpfl | Most lateral intersection of the frontal and parietal bones, taken on the parietal, left side | | | x | x | |
| lplpp | Most posterolateral point on the palatine plate, left side | | x | | | |
| lpsq | Most posterior point on the posterior extension of the forming squamosal, left side | | x | | | x |
| lpto | Most posteromedial point on the parietal, left side | | | x | x | x |
| lva | Most posterior point on the left ala of the vomer | | | | | x |
| raif | Most anteroinferior point on the frontal bone, right side | | | x | x | |
| ralf | Most anteromedial point on the frontal bone, right side | | | x | | |
| ralp | Most anterolateral point on the palatine plate, right side | | | | | |
| rasph | Posteromedial point of the inferior portion of the right alisphenoid | | | | | x |
| rflac | Intersection of frontal process of maxilla with frontal and lacrimal bones, right side | | | | | x |

*Table 3 continued on next page*

*Table 3 continued*

| Landmark description | | Anatomical region of interest | | | | |
|---|---|---|---|---|---|---|
| | | Bony skull landmarks for ages E15.5, E17.5, P0, and P2 | | | | |
| Landmark abbreviation | Landmark definition | Olfactory capsule landmark set used in EDMA of E15.5–P2 | Braincase floor landmark set used in EDMA of E15.5–P2 | Lateral wall and roof of pre-occipital and occipital region landmark set used in EDMA of E15.5–P2 | Lateral wall and roof of pre-occipital region landmark set used in Morphological Integration analysis | Global skull landmark set used in EDMA of E17.5, P0, and P2 |
| rfppm | Most supero-posterior point of the premaxilla accounting for the lateral part of the nasal aperture, right side | x | | | | x |
| riohd | Most distal point of the infraorbital hiatus, right side | x | | | | x |
| rmaxi | The midline point on the premaxilla between the incisor and the nasal cavity just anterior of the incisive foramen, right side | x | | | | x |
| rnasapl | Most supero-anterior point of the premaxilla accounting for the lateral part of the nasal aperture, right side | x | | | | x |
| roci | The supero posterior point on the ectocranial surface of occipital lateralis on the foramen magnum, right side | | x | x | | x |
| rpfl | Most lateral intersection of the frontal and parietal bones, located on the frontal, right side | | | x | x | |
| rplpp | Most posterolateral point on the palatine plate, right side | | x | | | |
| rpns | Most anterolateral indentation at the posterior edge of the palatine plate, right side | | | | | x |
| rpsq | Most posterior point on the posterior extension of the forming squamosal, right side | | x | | | x |
| rpto | Most posteromedial point on the parietal, right side | | | x | x | x |
| rva | Most posterior point on the right ala of the vomer | | | | | x |

remain wide relative to *Fgfr2c$^{+/+}$* littermates at P2 but are relatively reduced rostro-caudally (***Figure 5C and F***). Only bones of the most posterior aspect of the braincase floor remain relatively large in *Fgfr2c$^{C342Y/+}$* mice at P2 (***Figure 5C***). That the majority of *Fgfr2c$^{C342Y/+}$* skull dimensions are small relative to *Fgfr2c$^{+/+}$* littermates at P2 suggests that these differences are the result of altered early postnatal growth patterns in *Fgfr2c$^{C342Y/+}$* mice.

## Morphological integration of chondrocranium and dermatocranium

Morphological integration (MI) refers to patterns of correlation and/or covariation among organismal traits with the degree of integration measured by the intensity of statistical association in the phenotype. Patterns of covariation emerge because organisms are constructed of units or modules, which are coherent within themselves yet part of a larger unit. Modules result from structural or developmental associations within an organism (***Chernoff and Magwene, 1999***; ***Motch Perrine et al., 2017***; ***Olson and Miller, 1958***), but can also be outcomes of sample-specific developmental architecture

**Table 4.** Form difference of bony skulls.

Results (p values) of non-parametric null hypothesis tests for form differences euclidean distance matrix analysis (EDMA) of bony skull regions between *Fgfr2c$^{C342Y/+}$* mice and their *Fgfr2c$^{+/+}$* littermates using the expanded set of landmarks.

| Age | Olfactory capsule | Braincase floor | Lateral wall and roof of preoccipital and occipital region |
|---|---|---|---|
| Embryonic day 17.5 (E17.5) | 0.003 | 0.270 | 0.252 |
| Postnatal day 0 (P0) | 0.003 | 0.004 | 0.038 |
| Postnatal day 2 (P2) | 0.001 | 0.397 | 0.027 |

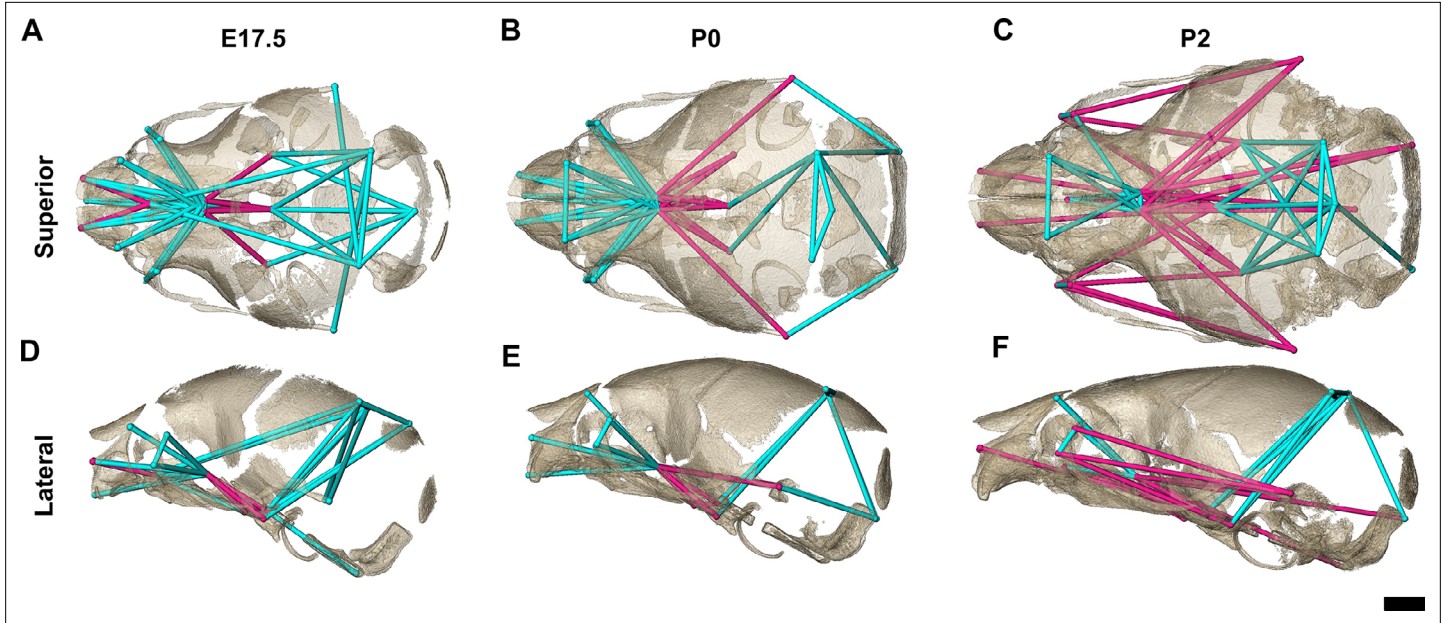

**Figure 5.** Euclidean distance matrix analysis of the bony skull during late prenatal and early postnatal stages. Increased mineralization allowed a larger set of landmarks to be used for statistical comparison of skull shape between genotypes at embryonic day 17.5 (E17.5), postnatal day 0 (P0), and P2 (as compared to *Figure 3*). Superior (**A–C**) and lateral (**D–F**) views of linear distances of the bony skull that are statistically significantly different between genotypes by confidence interval testing (α=0.10) shown on the dermatocranium of a *Fgfr2c$^{+/+}$* mouse at E17.5 (**A, D**), P0 (**B, E**), and P2 (**C, F**). Blue lines indicate linear distances that are significantly larger in *Fgfr2c$^{C342Y/+}$* mice; fuchsia lines indicate linear distances that are significantly reduced in *Fgfr2c$^{C342Y/+}$* mice. Quantitative patterns reveal a reversal in relative size postnatally, with the *Fgfr2c$^{C342Y/+}$* skull becoming generally smaller than skulls of *Fgfr2c$^{+/+}$* littermates. Scalebar = 1 mm.

and variation (*Hallgrímsson et al., 2009*) indicative of shared regulatory processes (*Carroll, 2001*; *Weiss, 2005*). We use a comparative study of MI of the chondrocranium and dermatocranium in *Fgfr2c$^{C342Y/+}$* mice and *Fgfr2c$^{+/+}$* littermates to determine whether coordinated patterns of association within and between these modules are altered by a Fgfr2 genetic variant.

Linear distances within the chondrocranium and dermatocranium were estimated from 3D coordinates of landmarks (*Table 1* and *Table 3*) and used to statistically compare MI patterns in *Fgfr2c$^{C342Y/+}$* and *Fgfr2c$^{+/+}$* mice within the chondrocranium, within the dermatocranium, and between chondrocranium and dermatocranium at E15.5 and E17.5 using previously published methods (*Richtsmeier et al., 2006*). MI patterns reported here are based on correlation matrices estimated using MIBoot, a Windows based software package (*Cole III TM, 2002a*) (correlation matrices estimated using MIboot available at DOI 10.26207/qgke-r185). We consider any correlation coefficient with value of 0.60 or greater as indicative of a relatively strong association, whether the correlation is positive or negative.

At E15.5, the mean of the absolute values of the correlation coefficients ($\bar{r}$) among chondrocranial dimensions is large in *Fgfr2c$^{C342Y/+}$* mice ($\bar{r}$ = 0.73) relative to *Fgfr2c$^{+/+}$* mice ($\bar{r}$ = 0.53) but the pattern

**Table 5.** Morphological integration of chondrocranium and dermatocranium.

Mean ($\bar{x}$) and standard deviation (s) of the absolute value of correlation coefficients for all chondrocranium measures, all dermatocranium measures, and between all chondrocranium and dermatocranium measures for embryonic day 15.5 (E15.5) and E17.5 samples used in analysis.

| Age | Genotype | Dermatocranium $\bar{x}$ | s | Chondrocranium $\bar{x}$ | s | Dermatocranium and Chondrocranium $\bar{x}$ | s |
|---|---|---|---|---|---|---|---|
| | Affected | 0.62 | 0.33 | 0.73 | 0.25 | 0.65 | 0.30 |
| E15.5 | Unaffected | 0.68 | 0.31 | 0.53 | 0.29 | 0.42 | 0.25 |
| | Affected | 0.59 | 0.29 | 0.61 | 0.28 | 0.46 | 0.26 |
| E17.5 | Unaffected | 0.52 | 0.28 | 0.47 | 0.28 | 0.49 | 0.27 |

of correlation is similar in the two samples with few (14%) correlations significantly different between the two genotypes (*Table 5*). By E17.5 the mean of the absolute values of the correlation coefficients have decreased in both samples but remain relatively high in *Fgfr2c*$^{C342Y/+}$ mice ($\bar{r}$ = 0.61) and the number of within-chondrocranial correlation coefficients that are significantly different between the samples is further reduced (9%). These results reveal a remarkable correspondence in overall patterns of within-chondrocranial associations in the two genotypes and a sustained increase in strength of the correlations in *Fgfr2c*$^{C342Y/+}$ mice relative to *Fgfr2c*$^{+/+}$ mice.

At E15.5, approximately one day after the initial mineralization of cranial dermal bone, the mean of the absolute values of correlation coefficients among dermatocranial dimensions are relatively strong in both genotypes (*Table 5*) and only 20 (9%) of the correlation coefficients among dermatocranial dimensions are significantly different between the two genotypes. By E17.5 the mean of the absolute value of correlation coefficients have decreased in both samples, though by a lesser amount in *Fgfr2c*$^{C342Y/+}$ mice, and a similarly small number of correlations are significantly different between genotypes.

Association of the chondrocranium and dermatocranium in *Fgfr2c*$^{C342Y/+}$ mice is strong ($\bar{r}$ = 0.65) relative to their *Fgfr2c*$^{+/+}$ littermates ($\bar{r}$ = 0.42) at E15.5 and statistical analysis of the difference in MI reveals 183 (41.5%) of the correlations to be significantly different between genotypes. Of these significant differences, 124 (28.1%) are due to a greater absolute magnitude of correlation in *Fgfr2c*$^{C342Y/+}$ mice relative to *Fgfr2c*$^{+/+}$ littermates while 59 (13.4%) of the differences are due to a significantly stronger association between chondrocranium and dermatocranium in *Fgfr2c*$^{+/+}$ littermates. At E15.5,

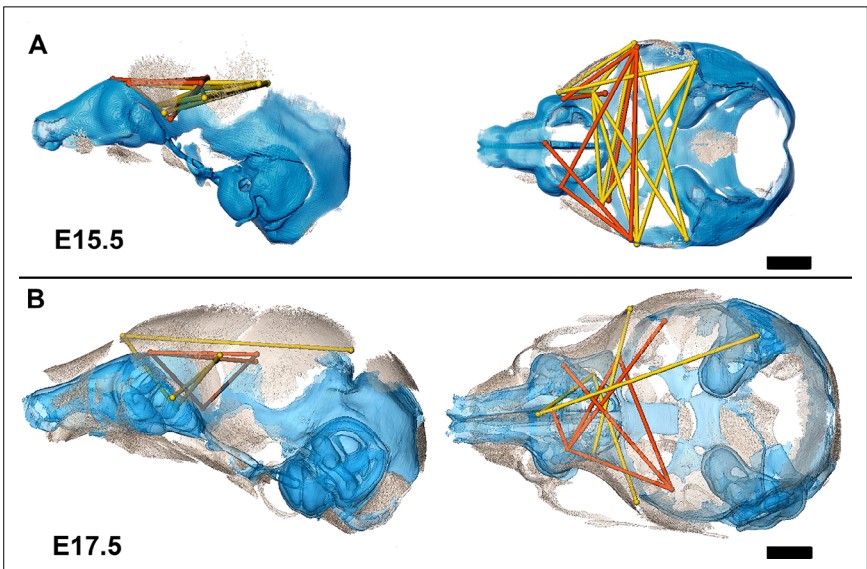

**Figure 6.** Summary of statistically significant differences in morphological integration of dermatocranium and chondrocranium between genotypes with two videos. (**A**) Linear distance pairs from the dermatocranium (yellow) and chondrocranium (orange) whose association is statistically stronger (α=0.10) in *Fgfr2c*$^{C342Y/+}$ mice relative to *Fgfr2c*$^{+/+}$ mice at embryonic day 15.5 (E15.5) and (**B**) at E17.5. Left lateral (at left) and superior (at right) views shown. Scalebars = 1 mm.

The online version of this article includes the following video for figure 6:

**Figure 6—video 1.** Three-dimensional reconstruction of the superimposed isosurfaces of an embryonic day 15.5 (E15.5) *Fgfr2c*$^{+/+}$ mouse chondrocranium and skull with linear distance pairs from the dermatocranium (yellow) and chondrocranium (orange) whose association is statistically stronger (α=0.10) in *Fgfr2c*$^{C342Y/+}$ mice relative to *Fgfr2c*$^{+/+}$ mice.

https://elifesciences.org/articles/76653/figures#fig6video1

**Figure 6—video 2.** Three-dimensional reconstruction of the superimposed isosurfaces of an embryonic day 17.5 (E17.5) *Fgfr2c*$^{+/+}$ mouse chondrocranium and skull with linear distance pairs from the dermatocranium (yellow) and chondrocranium (orange) whose association is statistically stronger (α=0.10) in *Fgfr2c*$^{C342Y/+}$ mice relative to *Fgfr2c*$^{+/+}$ mice.

https://elifesciences.org/articles/76653/figures#fig6video2

the significant differences in correlation patterns are of two general types: (1) correlations between specific chondrocranium and dermatocranium measures are moderately to strongly negative in $Fgfr2c^{+/+}$ littermates while being strongly positive in $Fgfr2c^{C342Y/+}$ mice indicating pairs of measures that vary in opposite directions in typically developing mice but that tend to increase (or decrease) jointly when the $Fgfr2$ variant is present; and (2) correlations that are moderately positive in $Fgfr2c^{+/+}$ mice and strongly negative in $Fgfr2c^{C342Y/+}$ mice describing relationships among chondrocranial and dermatocranial measures that are of low to medium positive intensity in typically developing mice but that vary strongly in opposite directions when the $Fgfr2$ variant is present (*Figure 6A*; *Figure 6—video 1*).

By E17.5, the lateral walls of the chondrocranium are dissolving as dermal bones mineralize and expand in size, and the mean association between the two modules decreases in $Fgfr2c^{C342Y/+}$ mice and increases slightly in $Fgfr2c^{+/+}$ mice yielding similar mean values between genotypes. The number of significant differences in correlations between dermatocranial and chondrocranial dimensions is reduced to 107 (24.3%) at E17.5 suggesting that the similar mean values are coupled with similar patterns of association between the two cranial modules at this age. Of these significant differences, 57 (12.9%) of them indicate relationships between specific chondrocranium and dermatocranium measures that are mildly to strongly negative in $Fgfr2c^{+/+}$ mice but mildly to strongly positive in $Fgfr2c^{C342Y/+}$ mice (*Figure 6B*; *Figure 6—video 2*), while 50 (11.3%) vary from mildly negative to strongly positive in $Fgfr2c^{+/+}$ mice but are moderately to strongly negative in $Fgfr2c^{C342Y/+}$ mice.

## Discussion

We have provided an improved method for segmentation and visualization of embryonic cranial cartilage by PTA-enhanced microCT imaging and used these data to reveal local and global variations of chondrocranial morphology and its relationship to the dermatocranium in mice carrying an Fgfr2 variant that is associated with Crouzon syndrome. Our detailed observations of chondrocranial morphology over embryonic time demonstrate the direct effects of the *Fgfr2c C342Y* variant on cartilage via chondroblast dysregulation resulting in malformation of the chondrocranium. $Fgfr2c^{C342Y/+}$ mice have a greater amount of cartilage and a chondrocranium that is generally larger and differently shaped relative to their $Fgfr2c^{+/+}$ littermates at every embryonic age studied. The dermatocranial elements of $Fgfr2c^{C342Y/+}$ mice form on the ectocranial surfaces of cartilage and match the contours and shapes of associated chondrocranial elements contributing to a generally larger and dysmorphic embryonic dermatocranium. Data support our hypothesis that the prenatal development of the chondrocranium and dermatocranium is integrated with the relationship contributing to skull morphogenesis, and suggest that while the chondrocranium is present, its morphology influences the formation and growth of dermatocranial elements.

Our findings have significant implications for understanding the role of embryonic cranial cartilage in the initial formation, configuration, and development of cranial dermal bone. Functional explanations for the chondrocranium are appropriate because no modern vertebrate has lost this skeleton during evolution. The ability of cartilage to grow interstitially and by accretion means that the cranial endoskeleton, unlike the cranial dermatoskeleton, can change shape dynamically during embryogenesis acting as a progressively transforming scaffold for developing dermal bone. The transient nature of the chondrocranium is one reason why so little is known about its role in craniofacial development and mouse models provide an ideal tool for addressing questions pertaining to its role in typical development, craniofacial disease, and potentially, evolution.

Craniosynostosis is a relatively common birth defect, second only to clefts of the lip and palate (*Heuzé et al., 2014*). Syndromes of Pfeiffer, Crouzon, Apert, Saethre-Chotzen, and Muenke comprise the most common of the FGFR-related craniosynostosis syndromes. Details of how the disease-associated genetic variants interrupt intracellular signaling is the focus of much research, but how those changes contribute to the assembly of disease phenotypes has received less attention. For example, it is not known if midfacial retrusion, a complex trait involving cartilage, bone, and soft tissues of the face and jaws and shared by most individuals with FGFR-related craniosynostosis syndromes, is produced by similar processes in patients carrying different FGFR variants. Mouse models that recapitulate the genetic basis for, and phenotypic consequences of, specific FGFR variants provide an experimental system to expand our knowledge of the production of FGFR-related phenotypes. Most of the work with craniosynostosis-associated genetic variants have focused on the

bony skull of mouse models for craniosynostosis, or on human cell lines to demonstrate how specific variants alter the processes of proliferation, differentiation, apoptosis, and/or polarity of osteoblast lineage cells as they differentiate. Exceptions include a study of *Fgfr2c^C342Y/C342Y^* mice suggesting that many phenotypic aberrations stem from a primary failure of mesenchymal condensation contributing to aberrant cartilage and bone (*Peskett et al., 2017*), observations of enhanced tracheal cartilage formation in *Fgfr2* mouse lines suggesting that abnormal chondrogenesis occurred (*Lam et al., 2021*; *Wang et al., 2005*), and studies that demonstrate cartilage-autonomous effects of Fgfr2 variants on the septum nasi and other facial cartilages (*Holmes et al., 2018*; *Kim et al., 2021*). *Holmes et al., 2018* found nasal cavity volume reduction and cartilage thickening to contribute significantly to the prenatal midface phenotype in two Apert syndrome mouse models (*Fgfr2^+/S252W^* and *Fgfr2^+/P253R^*) and the Crouzon mouse model used here, but that structural and cellular changes resulting in midfacial dysgenesis differ among specific *Fgfr2* variants. *Kim et al., 2021* found increased septal chondrocyte hypertrophy and thickening of the septum nasi postnatally to contribute to midfacial deformities in septum nasi-associated structures using a mouse line carrying a chondrocyte specific *Fgfr2 S252W* variant (Col2a1-cre; Fgfr2S252W/+). Both studies reveal midfacial dysgenesis in FGFR2-related craniosynostosis to be a complex phenotype arising from the combined effects of aberrant signaling in multiple craniofacial tissues including cartilage.

The chondrocrania of *Fgfr2c^C342Y/+^* Crouzon syndrome mice are composed of more and larger cartilage cells accompanied by more extracellular matrix, a finding consistent with the significantly larger size and increased thickness of their chondrocrania. This is the first demonstration that Fgf/Fgfr signaling directly affects chondrocranial shape through changes in chondrocyte characteristics contributing to the abnormal craniofacies of Crouzon syndrome. Mechanisms controlling the activity of chondrocytes in the identified developing regions of interest are multifaceted and time sensitive. Most instances of a significant increase in chondrocyte number, size, or matrix composition in *Fgfr2^+/+^* embryos can be directly associated with significant increase in other measures of chondrocyte and cartilage size in *Fgfr2^C342Y/+^* embryos at the same time. However, when this does not occur (e.g. matrix in braincase floor at E15.5, *Figure 3L*), it could potentially signal that statistically insignificant changes in other cellular characteristics have additive or interactive effects with biological significance that offsets the statistically significant increases identified in *Fgfr2c^+/+^* embryos. The morphology of the chondrocranium is the result of independent, integrated, and potentially additive effects of dynamic changes at the cellular level. As cartilages of the chondrocranium form individually, appearing at different points of embryonic time and maturing according to their own developmental schedule, the *Fgfr2c* variant may be affecting chondrocyte maturation and cell cycle differently across cartilages and within cartilage zones (e.g. proliferative and hypertrophic) such that an alternate approach to histological assessment is required.

Prenatal bony skulls of *Fgfr2c^C342Y/+^* mice are larger than those of their *Fgfr2c^+/+^* littermates, while skulls of adult *Fgfr2c^C342Y/+^* mice are relatively smaller with domed cranial vaults. Our analyses span prenatal and postnatal development revealing a transformative change in skull morphology and growth dynamics initiating late in prenatal development with disintegration of the transient chondrocranial cartilages. Our analyses highlight the significance of the cartilaginous scaffold to shapes of dermal bones, and advance embryonic cranial cartilage as a potential therapeutic target for craniofacial disease.

While it is known that the *Fgfr2c C342Y* variant results in constitutive activation of the receptor associated with up-regulation of osteoblast proliferation, our results reveal that this variant directly targets the chondrocyte lineage producing alterations in chondrocranial size and shape. The *Fgfr2c C342Y* variant produces change in chondrocyte size, chondrocyte number, and cartilage extracellular matrix area resulting in a morphologically distinct chondrocrania that indirectly influences prenatal dermatocranial element position, size, shape, and growth. The known regulatory effects on the osteoblast lineage may function at the cellular level prenatally but appear to direct the size and shape of forming dermal bone tissue differentially contingent on whether the chondrocranium is present (prenatally) or absent (postnatally). Once chondrocranial elements either disappear or mineralize endochondrally, size and shape of dermal bones begin transformations toward shapes seen in adult skulls. This suggests that the earliest dermal bone in *Fgfr2c^C342Y/+^* mice acts non-autonomously, in coordination with the variant's effects on chondrocytes. When chondrocytes of nearby cranial cartilages disappear; however, dermal bone behaves autonomously.

Of the three main hypotheses, we proposed to explain the relationship between chondrocranial cartilage and dermal bone formation, our results demonstrate that the *Fgfr2* variant affects the chondrocyte series and the osteoblast lineage and increases integration of chondrocranial and dermatocranial development prenatally. Studies of morphological integration (MI) reveal an elevated magnitude of association between chondrocranium and dermatocranium of *Fgfr2c*$^{C342Y/+}$ mice at E15.5 matching the results of previous analyses of the skulls of *Fgfr2*$^{+/S252W}$ and *Fgfr2*$^{+/P253R}$ Apert syndrome mouse models at P0 (*Martínez-Abadías et al., 2011*) that suggested FGF/FGFR signaling as a covariance-generating process in skull development acting to modulate MI intensity. The physical and developmental aspect of dermatocranium-chondrocranium integration is mirrored in reduced MI intensity between chondrocranium and dermatocranium for both genotypes at E17.5 as portions of the chondrocrania begin to dissolve.

Our findings are relevant to various fields and challenge traditional thinking about the role of cartilage in the formation of dermal bone. While the association of cartilage is well defined for endochondral ossification, intramembranous ossification is commonly described as mineralization that proceeds 'without a cartilaginous model'. Our data are the first to provide clear evidence of a developmental relationship between cartilaginous elements of the chondrocranium and bones of the dermatocranium. The combination of data presented here and elsewhere *Kawasaki and Richtsmeier, 2017a*; *Pitirri et al., 2020* demonstrates that these relationships underlie normal craniofacial development and dysmorphogenesis, and may offer a mechanistic explanation for the production of cranial variation across species, and even over evolutionary time. Our study supports the assertion that chondrocranial cartilages function as a scaffold, but also as a guide, significantly influencing the position, size, and shape of developing dermal bone. The relationship is temporary however and appears to diminish with the departure of transient cartilages, highlighting the critical, but fleeting impact of chondrocranial cartilage on dermal bone.

## Ideas and speculation

Our findings hold significance for the fields of 3D imaging, craniofacial development, disease, and evolution. The 3D reconstructions and visualizations of the two skeletal systems offer insightful views of little-known physical relationships that can aid in the formulation of functional hypotheses about the timing and emergent properties of neighboring cranial tissues. Our observations indicate a strong link between cranial cartilages and cranial dermal bone development, and it is likely that other genetic variants can affect the chondrocranium prior to mineralization of cranial bone. The evidence presented here of a relationship between the chondrocranium and dermatocranium advocates for a potential reassessment of the traditional definition of intramembranous ossification as a process that lacks any cartilage involvement.

In our example, it appears that the indirect effect of chondrocranial maldevelopment on dermal bone is physical or biomechanical and time sensitive as the relative size and shape of the bony skull of the two genotypes changes when the lateral walls of the chondrocranium break down. It is equally probably however that the chondrocranium-dermatocranium boundary functions as a signaling interface during normal craniofacial development. In typically developing mice the location of the coronal suture corresponds with, and may be predetermined by, the anterior edge of the TTR which is established as early as E13.5 (*Figure 1C*; *Figure 1—figure supplement 1*; *Figure 2*; *Figure 2—figure supplement 1*), much earlier than mineralization of the frontal and parietal bones (*Figure 2*). Osteoblasts destined to form the parietal bone do not differentiate rostral to the edge of the TTR in typically developing mice (*Kawasaki and Richtsmeier, 2017a*). As the lateral wall including the TTR shows significant changes in mice carrying the Fgfr2c C342Y variant, this boundary, and its role in formation of the coronal suture might be altered when the variant is present.

It is not uncommon for researchers to report 'ectopic' chondrocyte derived tissue in the study of craniofacial development and disease (e.g. *Bartoletti et al., 2020*; *He and Soriano, 2017*; *Holmes and Basilico, 2012*). Although the chondrocranial changes identified in *Fgfr2c*$^{C342Y/+}$ mice are ectopic in the sense that they are located 'in an abnormal place or position,' future studies should distinguish between the effect of genetic variants on the size, shape, and position of typically developing chondrocranial cartilages and effects that cause novel cartilages to form in locations where cranial cartilage is not normally found. Truly ectopic cartilage may not have a tight link with dermal bone formation and such distinctions could be predictors of emerging craniofacial (dys)morphology.

Finally, our demonstration that the development of the chondrocranium and dermatocranium is integrated may not be limited to mouse development but could denote an evolutionary mechanism of vertebrate skull integrity. Though in our experience the relationship between specific chondrocranial cartilages and dermal bones is constant across mouse strains, there exist interspecies differences in the cartilages that compose the chondrocranium (*de Beer, 1937*), and the association of chondrocranial elements with specific dermal bones varies over time and across species. Some cartilages of the mouse chondrocranium are not present in humans for example (*Kawasaki and Richtsmeier, 2017a*), and their function is most likely assumed by an alternate cartilage. Historic works by *de Beer, 1937*; *Moore, 1981*; *Starck, 1979* and contemporary works (e.g. *Werneburg, 2020*) provide information on the incredible variation of chondrocranial morphology across mammals and vertebrates. Though the link between the chondrocranium and dermatocranium is robust, the association between the two skeletal systems appears to have the ability to vary and can evolve, with the potential for differing signaling systems to direct these links in different species.

# Materials and methods

## Key resources table

| Reagent type (species) or resource | Designation | Source or reference | Identifiers | Additional information |
|---|---|---|---|---|
| Strain, strain background (*Mus musculus*, CD1) | *Fgfr2c^{C342Y/+}* | *Eswarakumar et al., 2004* | | Laboratory of Dr. Richtsmeier (Pennsylvania State University); craniosynostosis mouse model on a CD1 background |
| Software, algorithm | Avizo | ThermoFisher Scientific | RRID:SCR_014431 | https://www.thermofisher.com/us/en/home/electron-microscopy/products/software-em-3d-vis/avizo-software.html |
| Software, algorithm | Code for automatic chondrocranium segmentation with very sparse annotation via uncertainty-guided self-training | *Zheng et al., 2020*. https://doi.org/10.1007/978-3-030-59710-8_78 | | https://github.com/ndcse-medical/CartSeg_UGST |
| Software, algorithm | Euclidean Distance Matrix Analysis (EDMA) | *Lele and Richtsmeier, 2001*; ISBN-13: 978–0849303197 ISBN-10: 0849303192 | | https://getahead.la.psu.edu/resources/edma/ and https://github.com/psolymos/EDMAinR; *Solymos, 2021* |
| Software, algorithm | IBM SPSS Statistics | IBM | SCR_016479 | https://www.ibm.com/products/spss-statistics |
| Software, algorithm | Statistical Analysis System (SAS) | SAS | RRID:SCR_008567 | http://www.sas.com |
| Other | Weigert's Iron Hematoxylin | Sigma | HT1079 | Per manufacturer's protocol, 1:1 solution Parts A:B http://www.ihcworld.com/_protocols/special_stains/safranin_o.htm |
| Other | Safranin O | Sigma-Aldrich | 115980025 | 0.1% solution with distilled water http://www.ihcworld.com/_protocols/special_stains/safranin_o.htm |
| Other | Fast Green FCF | Sigma-Aldrich | F7252 | 0.05% solution with distilled water http://www.ihcworld.com/_protocols/special_stains/safranin_o.htm |
| Other | Acetic Acid | Fisher | A38SI-212 | 1% solution with distilled water http://www.ihcworld.com/_protocols/special_stains/safranin_o.htm |

## Sample

Mice were produced, sacrificed, and processed in compliance with animal welfare guidelines approved by the Pennsylvania State University Animal Care and Use Committee (#46558). Based upon timed mating and evidence of pregnancy, litters were sacrificed and collected as appropriate (See *Table 6* for sample sizes for specific analyses). Mice were housed in conventional cages (plastic rectangular tank; up to five adults) and placed in individually ventilated racks with corncob bedding, 12:12 hr light:dark cycle, ad libitum food and water access, environmental enrichment including nesting shredded paper and plastic toys. The bed is changed once a week. Mice were assessed daily for illness or injury. PTA staining, alizarin red, and alcian blue staining were performed as previously described (*Behringer et al., 2014*; *Lesciotto et al., 2020*).

## Imaging protocols

MicroCT images for bone and PTA-enhanced (PTA-e) microCT images for soft tissue analyses were acquired by the Center for Quantitative Imaging at the Pennsylvania State University (http://www.cqi. psu.edu/) using the General Electric v|tom|x L300 nano/microCT system. This is a dual-tube system with a 300-kV microfocus tube for larger specimens and a 180-kV nanofocus tube for smaller specimens. Although specimens may be scanned using either tube, we found the greatest resolution and scan quality were typically produced by the 180-kV tube for embryonic specimens and the 300-kV tube for postnatal specimens. Image data were reconstructed on a 2024 × 2024 pixel grid as a 32-bit volume but may be reduced to 16-bit volume for image analysis using Avizo 2020.2 (ThermoFisher Scientific, Waltham, MA). Scanning parameters varied from 60 to 100 kV and 75–170 µA, to accommodate age group and type of scan performed. Voxel sizes ranged from 6.9 to 15 µm for bone scans and 4.5–8 µm for PTA-e scans.

## Data collection

### Segmentation of bone

A hydroxy apatite (HA) bone phantom was included alongside specimens being scanned for bone. A minimum threshold of 70–100 mg/cm$^3$ partial density HA was used to reconstruct bony isosurfaces in Avizo 2020.2. Data were passed through a median filter to remove noise and the Volume Edit tool of Avizo was used to remove any material not part of the skull. Specific bone volumes were determined using the Material Statistics module of Avizo by researchers blinded to genotype. Bone volumes were compared between $Fgfr2c^{C342Y/+}$ mice and $Fgfr2c^{+/+}$ littermates in IBM SPSS 25 software (IBM, Armonk, NY) using non-parametric Mann-Whitney U tests due to violations of assumptions of homogeneity or variance and/or normality. Following bone volume measurement, 3D isosurfaces were compacted to 1,000,000 faces each in the Simplification Editor of Avizo 2020.2 prior to landmarking.

### Segmentation of embryonic cartilage

We previously reported an automatic deep learning based chondrocranium segmentation approach (*Zheng et al., 2020*). Although deep learning based FCNs have achieved great successes on both generic and biomedical image segmentation (*Long et al., 2015*; *Ronneberger et al., 2015*; *Zheng*

**Table 6.** Sample sizes of embryonic mice used in analyses.

Specimen matched bone and phosphotungstic acid enhanced (PTA-e) scans were used for morphological integration (MI) analysis.

| Age | Genotype | Bone Scan | | | PTA Scan | MI | Histology |
|---|---|---|---|---|---|---|---|
| | | E15.5, E16.5, E17.5 EDMA | E17.5, P0, P2 EDMA | E17.5, P0 Bone volumes | | | |
| E13.5 | $Fgfr2c^{+/+}$ | 0 | 0 | 0 | 3 | 0 | 4 |
| | $Fgfr2c^{C342Y/+}$ | 0 | 0 | 0 | 3 | 0 | 4 |
| E14.5 | $Fgfr2c^{+/+}$ | 0 | 0 | 0 | 5 | 0 | 7 |
| | $Fgfr2c^{C342Y/+}$ | 0 | 0 | 0 | 5 | 0 | 7 |
| E15.5 | $Fgfr2c^{+/+}$ | 7 | 0 | 0 | 5 | 5 | 6 |
| | $Fgfr2c^{C342Y/+}$ | 4 | 0 | 0 | 4 | 4 | 6 |
| E16.5 | $Fgfr2c^{+/+}$ | 7 | 0 | 0 | 5 | 0 | 6 |
| | $Fgfr2c^{C342Y/+}$ | 7 | 0 | 0 | 5 | 0 | 5 |
| E17.5 | $Fgfr2c^{+/+}$ | 13 | 31 | 13 | 5 | 5 | 4 |
| | $Fgfr2c^{C342Y/+}$ | 13 | 18 | 14 | 5 | 5 | 5 |
| P0 | $Fgfr2c^{+/+}$ | 0 | 11 | 10 | 0 | 0 | 0 |
| | $Fgfr2c^{C342Y/+}$ | 0 | 11 | 10 | 0 | 0 | 0 |
| P2 | $Fgfr2c^{+/+}$ | 0 | 13 | 0 | 0 | 0 | 0 |
| | $Fgfr2c^{C342Y/+}$ | 0 | 16 | 0 | 0 | 0 | 0 |

*et al., 2019*), segmenting chondrocrania in 3D microCT images remains a very challenging task. Due to high difficulty in labeling complicated objects (embryonic cranial cartilages) in large 3D microCT images to provide sufficient training data for deep learning model training, we must resort to sparse annotation (i.e. labeling only a very small subset of 2D slices in the training set of whole 3D volumes) for training our 3D segmentation model, while still enabling our model to segment the unseen whole volumes (including the delicate and detailed ROIs) with good accuracy. To this end, we developed a new, two-phase approach: (1) automatically segmenting the majority of the chondrocranium with very sparse annotation performed by experts in anatomy that bridges the performance gap compared with full annotation; (2) integrating limited human corrections to fine-tune the model. We present a high-level description of our approach below.

1. Automatic chondrocranium segmentation with very sparse annotation via uncertainty-guided self-training. Manual annotation was performed by experts using Avizo 2020.2 (ThermoFisher Scientific, Waltham, MA). We started with selecting a very sparse subset of 2D slices (e.g. 2–10%) for annotation that represents and covers the unannotated slices of the whole training volumes well. We then used the annotated slices to train a judiciously designed K-head FCN to predict pseudo-labels (PLs) in the unannotated slices of the training volumes (for bridging the spatial annotation gap) as well as compute the associated uncertainty maps of the PLs (which quantify the pixel-wise prediction confidence or uncertainty). Guided by the uncertainty, we iteratively trained the FCN with PLs and improved its generalization ability in unseen volumes. Moreover, we integrated the segmentation results along three orthogonal planes to further boost the segmentation performance via ensemble learning. Experimental results showed that our approach achieves average Dice scores of 87% and 83% in the training and unseen (test) volumes, respectively, with only 3% annotation of the slices in the training volumes. More details of our approach and validations can be found in *Zheng et al., 2020*.

2. Model fine-tuning via human-aided corrections. The automatic segmentation accuracy in the first phase on extremely difficult ROIs (e.g. Meckel's cartilage and cranial vault) may still not meet the requirement of quantitative analysis, because the model's generalizability is constrained by the highly sparse annotation and the unbalanced amounts of training pixels between easy and difficult regions. Hence, we first evaluated the inadequately segmented regions and manually corrected the algorithm-generated predictions, and then combined the annotations thus obtained and PLs to further fine-tune our segmentation model. This process did not incur too much computational costs. Consequently, most specimens were segmented almost perfectly by our model, except for extremely thin, small, or ambiguous regions in certain specimens. Finally, we manually corrected these local errors to generate an accurate chondrocranium model for quantitative analysis.

## Landmark data

Three dimensional coordinates of biologically relevant landmarks were collected from slices and isosurfaces created from microCT images of the specimens using Avizo 2020.2 (ThermoFisher Scientific, Waltham, MA). Specimens were digitized twice by the same observer, who was blinded to genotype, checked, and corrected for any gross error. Measurement error was minimized by averaging the coordinates of the two trials. A maximum of 5% error in landmark placement was accepted. *Table 1* and *Table 3* provide anatomical definitions of all landmarks used. Further information on landmark data can be found at https://getahead.la.psu.edu/landmarks/.

## Suture patency

Researchers blinded to genotype scored patterns of suture patency as visualized on microCT images for the coronal suture and three facial sutures in each mouse assigning qualitative scores of open, partially open, or fused to the entire length of the sutures using previously published protocols (*Motch Perrine et al., 2014*). These observations were used to show the relationship of suture patency patterns and craniofacial shape in both genotypes from E17.5to P2 (*Figure 4*).

## Statistical analyses

### Morphological comparison of embryonic cranial cartilage and bone

To statistically determine shape differences between groups, we used EDMA (*Lele and Richtsmeier, 2001*; *Lele and Richtsmeier, 1995*). EDMA converts 3D landmark data into a matrix of all possible

linear distances between unique landmark pairs and tests for statistical significance of differences between shapes using a boot-strapped hypothesis testing procedure and non-parametric boot-strapped confidence intervals. We used subsets of landmarks representing various anatomical regions to test for morphological differences of the nasal capsule, lateral walls, and braincase floor of the chondrocranium and the bony skull of $Fgfr2c^{C342Y/+}$ and $Fgfr2c^{+/+}$ mice. Use of these subsets in the evaluation of regional shape differences was done to bring the sample size closer to the number of landmarks considered for statistical testing. Significant differences of specific linear distances are evaluated by a 90% confidence interval produced through a non-parametric bootstrapping procedure (*Lele and Richtsmeier, 1995*). Rejection of the null hypothesis of similarity for linear distances enables localization of differences to specific dimensions. EDMA analyses were performed using WinEDMA (University of Missouri-Kansas City, Kansas City, MO),(*Cole III TM, 2002b*) and EDMAinR (University of Alberta, Edmonton, Canada) (*Solymos, 2021*).

## Principal components analysis of form

Ontogenetic variation in skull shapes were assessed using principal components analysis (PCA). To assess form (size and shape), all inter-landmark distances were *ln*-transformed and their variance-covariance matrix was used as the basis for the PCA (*Motch Perrine et al., 2014*). The amount of variation due to form is the sum of the variances for all of the *ln*-transformed linear measurements. All PCA were performed using SAS 9.4 (SAS Institute, Cary, NC). We scored suture patency separately (described above) and coded specimens in the PCA plot according to suture patency (*Figure 4*).

## Morphological integration

Though there are many methods to test hypotheses of cranial integration estimated using matrix correlations and/or covariances, here, we study integration within the chondrocranium, within the dermatocranium (excluding any landmarks on endochondral skull bones), and between the chondrocranium and dermatocranium. To avoid the use of superimposition when estimating correlation/covariance among traits and differences in these patterns, we use linear distances estimated from 3D coordinate locations of biological landmarks (*Richtsmeier et al., 2006*). The use of linear distances also circumvents the affine registration (a mapping that includes three translations, three rotations, three scales, and three shears) required to register data from microCT skull images and PTA-e microCT chondrocranial images.

   Our analysis provides information about how typical integration of chondrocranium and dermatocranium is altered in the presence of craniosynostosis-associated variants by statistically comparing patterns of correlation/covariance in $Fgfr2c^{C342Y/+}$ embryos and $Fgfr2c^{+/+}$ littermates using a previously published method (*Motch Perrine et al., 2017*; *Richtsmeier et al., 2006*). To statistically compare patterns of MI between genotypes we used a boot-strap based method (*Cole III TM, Lele S, 2002*; *Richtsmeier et al., 2006*) implemented in MIBoot (University of Missouri-Kansas City, Kansas City, MO), a Windows-based software package (*Cole III TM, 2002b*). 3D coordinates of 7 dermatocranial landmarks and 7 chondrocranial landmarks (see *Table 3* and *Table 1*) recorded from microCT and PTA-e microCT images, respectively, were used to estimate a total of 861 linear measures (42 unique linear distances among landmarks located on the dermatocranium and 42 unique linear distances estimated between chondrocranial landmarks) that were used in analysis. Within each age group, for each sample, a correlation/covariance matrix was estimated for unique linear distances pairs and a correlation difference matrix was estimated by subtracting the elements of the correlation matrix estimated for the $Fgfr2c^{C342Y/+}$ sample from the corresponding elements of the matrix estimated for the $Fgfr2c^{+/+}$ sample. If the correlation matrices are the same for two samples, then the correlation-difference matrix consists of zeros. If they are not similar, each element of the correlation difference matrix is statistically evaluated using a nonparametric bootstrap approach to estimate confidence intervals ($\alpha$=0.10). If a confidence interval does not include zero (the expected value under the null hypothesis of similarity), then the null hypothesis of similar associations for that linear distance pair is rejected. Using this method, we statistically compared the correlation patterns within the dermatocranium, within the chondrocranium, and between the dermatocranium and chondrocranium for $Fgfr2c^{C342Y/+}$ Crouzon syndrome mice and $Fgfr2c^{+/+}$ littermates at E15.5 and E17.5.

## Histology

Randomly selected specimen per age and genotype were labeled to conceal genotype, fixed overnight in 4% paraformaldehyde, processed for paraffin-based histology per standard protocol, serially sectioned at 7 µm using a manual rotary microtome, stained according to standard safranin O staining protocol, and imaged using Leica BX50 microscope, DFC450 camera, and LAS-X x-y scanning imaging software (Leica Biosystems, Allendale, NJ). Regions of interest stained with safranin O were identified and analyzed using Image-J color deconvolution and masks to count stained areas by color (Purple = nuclei, Orange = Cartilage matrix). Image files were labeled as to blind the investigator to the genotype of the specimen. At least three images were measured per region per individual (See *Table 6* for n). Non-parametric Mann-Whitney U tests were used to compare genotypes at each age in SPSS 25 software (IBM, Armock, NY) as there were violations of assumptions of homogeneity of variance and/or normality.

## Acknowledgements

The authors extend their gratitude to the staff at the Pennsylvania State University Center for Quantitative Imaging (https://iee.psu.edu/labs/center-quantitative-imaging) for production of excellent quality images used in this study and to Talia Pankratz Connell for assistance in image preparation. This research was supported in part by NICHD/NIH P01HD078233 and NIDCR/NIH R01 DE027677, R01DE031439, and NSF CCF-1617735.

## Additional information

### Funding

| Funder | Grant reference number | Author |
|---|---|---|
| National Institute of Dental and Craniofacial Research | R01DE027677 | Joan T Richtsmeier |
| Eunice Kennedy Shriver National Institute of Child Health and Human Development | P01HD078233 | Joan T Richtsmeier |
| National Institute of Dental and Craniofacial Research | R01 DE031439 | Joan T Richtsmeier |
| National Science Foundation | CCF-1617735 | Danny Z Chen |

The funders had no role in study design, data collection and interpretation, or the decision to submit the work for publication.

### Author contributions

Susan M Motch Perrine, Data curation, Formal analysis, Investigation, Project administration, Supervision, Validation, Visualization, Writing - original draft, Writing – review and editing; M Kathleen Pitirri, Data curation, Formal analysis, Investigation, Visualization, Writing – review and editing; Emily L Durham, Data curation, Formal analysis, Investigation, Validation, Visualization, Writing – review and editing; Mizuho Kawasaki, Investigation, Resources; Hao Zheng, Data curation, Methodology, Software, Validation, Visualization, Writing – review and editing; Danny Z Chen, Data curation, Funding acquisition, Methodology, Resources, Software, Supervision, Validation, Visualization, Writing – review and editing; Kazuhiko Kawasaki, Conceptualization, Writing – review and editing; Joan T Richtsmeier, Conceptualization, Formal analysis, Funding acquisition, Methodology, Project administration, Resources, Supervision, Validation, Writing - original draft, Writing – review and editing

### Author ORCIDs

Susan M Motch Perrine (iD) http://orcid.org/0000-0003-3412-221X
Emily L Durham (iD) http://orcid.org/0000-0002-6322-9393
Hao Zheng (iD) http://orcid.org/0000-0002-9790-7607

Danny Z Chen http://orcid.org/0000-0001-6565-2884
Kazuhiko Kawasaki http://orcid.org/0000-0003-1090-5340
Joan T Richtsmeier http://orcid.org/0000-0002-0239-5822

## Ethics

This study was performed in strict accordance with the recommendations in the Guide for the Care and Use of Laboratory Animals of the National Institutes of Health. All of the animals were handled according to approved institutional animal care and use committee (IACUC) protocols (#46558) of the Pennsylvania State University.

## Decision letter and Author response

Decision letter https://doi.org/10.7554/eLife.76653.sa1
Author response https://doi.org/10.7554/eLife.76653.sa2

## Additional files

### Supplementary files

• Transparent reporting form

### Data availability

Data have been made available through Penn State University Libraries ScholarSphere repository at https://doi.org/10.26207/qgke-r185 and include: bone micro-CT images, PTA-e micro-CT images, 3D reconstruction examples of the chondrocrania of one unaffected ($Fgfr2c^{+/+}$) and one affected ($Fgfr2c^{C342Y/+}$) at E13.5, E14.5, E15.5, E16.5, and E17.5, bone volumes, histological images, histomorphometric data, 3D landmark coordinate data, correlation matrices estimated by MIBoot used in MI analyses, PCA output, and suture scores. An interactive viewer for reconstructions of chondrocrania of mice carrying the Fgfr2 mutation and unaffected littermates is available on FaceBase (https://doi.org/10.25550/J-RHCA). Information on how to download the WinEDMA programs can be found at https://getahead.la.psu.edu/resources/edma and the EDMAinR programs are available on github (https://github.com/psolymos/EDMAinR, copy archived at swh:1:rev:a8d9efa10578250dc7a6b-6b993a6d85da1419ced). Code for automatic chondrocranium segmentation with very sparse annotation via uncertainty-guided self-training is available through https://github.com/ndcse-medical/CartSeg_UGST, copy archived at swh:1:rev:b4d901ccc477bb69ad8edcdf3f1503a0cb4a6405. PTA-e staining protocols for various embryonic ages of mice are available: https://doi.org/10.1002/dvdy.136.

The following dataset was generated:

| Author(s) | Year | Dataset title | Dataset URL | Database and Identifier |
|---|---|---|---|---|
| Richtsmeier JT, Motch Perrine SM, Pitirri MK | 2022 | A dysmorphic mouse model reveals developmental interactions of chondrocranium and dermatocranium | https://doi.org/10.26207/qgke-r185 | The Pennsylvania State University ScholarSphere, 10.26207/qgke-r185 |

The following previously published dataset was used:

| Author(s) | Year | Dataset title | Dataset URL | Database and Identifier |
|---|---|---|---|---|
| Chen D, Kawasaki K, Perrine SM, Pitirri M, Richtsmeier J, Zheng H | 2022 | 3D reconstructions of murine chondrocrania | https://doi.org/10.25550/J-RHCA | FaceBase, 10.25550/J-RHCA |

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
