## [Editor Report]

Richtsmeier and colleagues demonstrate that chondrocranium and dermatocranium development are associated and that mutations in Fgfr significantly alter skull shape in part via the chondrocranium by means of a 3D modeling technique. The study is inspiring in providing new data regarding the role of the chondrocranium in normal craniofacial development and shedding light on the putative correspondence between chondrocranial elements and dermal skull bones. This work will be of interest to readers in the fields of vertebrate developmental biology, evolutionary anatomy, genetic disease, and vertebrate paleontology.

---

## [Decision Letter]

**Decision letter after peer review:**

Thank you for submitting your article "Untangling over 400 million years of cooperation between chondrocranium and dermatocranium development" for consideration by eLife. Your article has been reviewed by 3 peer reviewers, including Min Zhu as the Reviewing Editor and Reviewer #1, and the evaluation has been overseen by Kathryn Cheah as the Senior Editor.

Essential revisions:

1) This manuscript suffers from the data transparency to support the conclusions.

2) The study in the current form fails to study the "400 million years of cooperation between chondrocranium and dermatocranium development" but does study the association between chondrocranium and dermatocranium development in one selected Crouzon model. The title and the conclusion go too much beyond the results they have gathered. It needs to be clearly discussed why the cranial suture model is suitable for discussing the association between the chondrocranium and dermatocranium and why that untangles 400 million years of cooperation between chondrocranium and dermatocranium development. In the current form, this is just a descriptive study of the Curzon mouse and does not provide innovative insight into the "evolution" of the chondrocranium and dermatocranium.

3) The linking between chondrocyte expression of the mutant Fgfr2 allele and Apert's syndrome should be clarified. One reviewer suggests that the potential issues from a disease perspective might be points to be addressed.

*Reviewer #1 (Recommendations for the authors):*

My concern is about the minimum age of cooperation between chondrocranium and dermatocranium development (title, L.33, L.53).

The age in Line 598 (470 million years ago or earlier) is correct. The authors can refer to Sansom and Andreev (2019). The earliest evidence of the dermatocranium is of Floin, Early Ordovician, over 470 Ma.

I. J. Sansom and P. S. Andreev, The Ordovician Enigma: Fish, First Appearances and Phylogenetic Controversies. In: Evolution and Development of Fishes, edited by Z. Johanson, C. Underwood and M. Richter, Cambridge University Press 2019.

Another relevant comment is about the sentence in Lines 57-59. The cranial dermal bone formation first evolved in 'ostracoderms' or jawless stem-Gnathostomata, then evolved in 'placoderms' or jawed stem-Gnathostomata, regressed in Chondrichthyes and retained in Osteichthyes. The authors can refer to Zhu et al. (2013) for the homology of dermal cranial bones across major vertebrate groups.

M. Zhu et al., 2013. A Silurian placoderm with osteichthyan-like marginal jaw bones. Nature, 502, 188-193.

*Reviewer #2 (Recommendations for the authors):*

My concerns with this work are exclusively limited to data presentation and data clarification.

It is very difficult to appreciate the changes referred to in the text and shown in Figures 1 and 2. Both of these figures should be annotated so that the elements and changes being discussed at each stage are readily observable in the figures.

The changes in cellular parameters in the chondrocranium are not intuitively understood. For example, what explains the decrease in chondrocyte number at e15.5 in the mutants? It appears to greatly increase again at e16.5. There are other examples in this figure that are also confusing. Can these dynamic changes be explained biologically, or are they a result of the sampling?

*Reviewer #3 (Recommendations for the authors):*

Bony fish, amphibians, lizards, birds, and other mammals should be included to address the 400 million years of cooperation between chondrocranium and dermatocranium.

---

## [Author Response]

Essential revisions:1) This manuscript suffers from the data transparency to support the conclusions.

We realize that *eLife* strongly encourages making all data available prior to article submission and requires all data be available before publication. Given our preliminary discussions with FaceBase and assignment of a doi for our data, we thought that the CT data would be uploaded by the time of review. However, making the image files available proved to be more difficult than anticipated due to their size (10-15 GB per scan), and even uploading the 3D reconstructions, which are much smaller than the data, had its limitations as we had to decimate the data for upload to FaceBase, resulting in valuable detail being lost. We have found success with Scholarsphere, a repository hosted by the Pennsylvania State University Libraries. All data are available at a single address: DOI 10.26207/qgke-r185 and include: all bone micro-CT images, all PTA-enhanced micro-CT images, all bone volumes, all histological images, histomorphometric data, all 3D landmark coordinate data, correlation matrices estimated by MIboot for MI analyses, details of PCA output, and all suture scores. Although FaceBase was unable to accommodate our large data sets, they worked with us to enable sharing reconstructions of chondrocrania of mice carrying the *Fgfr2* mutation and unaffected littermates in an interactive viewer (https://doi.org/10.25550/J-RHCA).

2) The study in the current form fails to study the "400 million years of cooperation between chondrocranium and dermatocranium development" but does study the association between chondrocranium and dermatocranium development in one selected Crouzon model. The title and the conclusion go too much beyond the results they have gathered. It needs to be clearly discussed why the cranial suture model is suitable for discussing the association between the chondrocranium and dermatocranium and why that untangles 400 million years of cooperation between chondrocranium and dermatocranium development. In the current form, this is just a descriptive study of the Curzon mouse and does not provide innovative insight into the "evolution" of the chondrocranium and dermatocranium.

We agree with this critique and realize that we distracted reviewers from our major message by focusing our title and some of our discussion on the evolution of the cranial endoskeleton and exoskeleton. We have changed our title to focus on the influence of growth of the chondrocranium on the dermatocranium as revealed by use of the *Fgfr2* mouse model. Beyond changing the title, we have corrected and shortened introductory statements, though we have kept short passages in the Introduction to refer to the separate evolution of the chondrocranium and dermatocranium, making corrections suggested by Reviewer 1 (see especially lines 56-67). We choose to keep this information in the manuscript because we feel that the separate evolution of these two structures and their eventual interaction during development in modern vertebrates is a fact worth mentioning that is unappreciated by most developmental biologists. We have made a concerted effort to refocus our message by reducing the mention of evolution and focusing on the impact of changes in the chondrocranium on the formation of the bony skull in the Discussion section of our resubmission. We have referred to evolutionary considerations in a few final sentences of the Ideas and Speculation section (lines 686-699).

Reference to the *Fgfr2* mouse model for Crouzon syndrome as a “cranial suture model” in reviews indicates that we were unsuccessful in communicating the value of these experimental models in understanding generalized craniofacial development, rather than limiting their usefulness to the study of suture patency and premature suture closure. Over the past 15 years, our group has used various *Fgfr* mouse models to reveal that craniosynostosis is more than a cranial suture disease: the FGFR craniosysnostosis syndromes represent growth disorders that affect nearly all craniofacial tissues (see especially https://doi.org/10.1002/wdev.227; https://doi.org/10.1242/dev.166488; https://doi.org/10.1002/dvdy.23903 ). The submitted paper builds on this body of work providing further evidence of this conclusion. But the submitted work is novel in showing that this specific *Fgfr2* mutation affects cartilage cellular processes resulting in quantifiable change in the composite, 3D, cartilaginous structure of the chondrocranium, a structure never fully studied in embryonic mice. Moreover, our work has also revealed for the first time, the impact of a malformed chondrocranium on dermatocranial morphology (as summarized in the Public Evaluation Summary). We have rewritten the manuscript to make clear that this is not only a study of suture patency, while explaining that our observations of changes of the chondrocranium in Fgfr2 Crouzon mice may have a role in premature suture closure (see lines 664-676).

3) The linking between chondrocyte expression of the mutant Fgfr2 allele and Apert's syndrome should be clarified. One reviewer suggests that the potential issues from a disease perspective might be points to be addressed.

Syndromes of Pfeiffer, Crouzon, Apert, Saethre-Chotzen, and Muenke comprise the most common of the FGFR-related craniosynostosis syndromes. Numerous variants found on the different members of the Fgfr family of genes are associated with these diseases. Though individuals with FGFR-related craniosynostosis syndromes share certain phenotypic features, each FGFR-related craniosynostosis syndrome is a complex genetic disorder, where the phenotypic manifestations of a specific genetic variant in each individual are products of genetic, environmental, and stochastic influences. A study by our group (https://doi.org/10.1242/dev.166488) used three *Fgfr2*-related craniosynostosis mouse models to demonstrate that structural and cellular changes resulting in midfacial dysgenesis differ between specific *Fgfr2* mutations. From this result, we reasoned originally that it could obscure rather than enlighten to refer to the impact of other *Fgfr2* mutations (in addition to Fgfr2c studied here) on cartilage development within the context of the current study. We respect the reviewer’s concern however. To offer a fuller picture of the complexities of the various Fgfr2 mutations on cartilage formation we have expanded that discussion to include what we think is the article to which Reviewer 3 referred (https://www.nature.com/articles/s41598-021-87260-5) (see especially lines 571-586).

Reviewer #1 (Recommendations for the authors):My concern is about the minimum age of cooperation between chondrocranium and dermatocranium development (title, L.33, L.53).The age in Line 598 (470 million years ago or earlier) is correct. The authors can refer to Sansom and Andreev (2019). The earliest evidence of the dermatocranium is of Floin, Early Ordovician, over 470 Ma.I. J. Sansom and P. S. Andreev, The Ordovician Enigma: Fish, First Appearances and Phylogenetic Controversies. In: Evolution and Development of Fishes, edited by Z. Johanson, C. Underwood and M. Richter, Cambridge University Press 2019.

We thank this reviewer for noting the inconsistencies in our writing and have made the suggested corrections in the text (Title is changed, lines 32-33, 55-67)

Another relevant comment is about the sentence in Lines 57-59. The cranial dermal bone formation first evolved in 'ostracoderms' or jawless stem-Gnathostomata, then evolved in 'placoderms' or jawed stem-Gnathostomata, regressed in Chondrichthyes and retained in Osteichthyes. The authors can refer to Zhu et al. (2013) for the homology of dermal cranial bones across major vertebrate groups.M. Zhu et al., 2013. A Silurian placoderm with osteichthyan-like marginal jaw bones. Nature, 502, 188-193.

Thank you for pointing out this error. These corrections have been made (see lines 61-64)

Reviewer #2 (Recommendations for the authors):My concerns with this work are exclusively limited to data presentation and data clarification.It is very difficult to appreciate the changes referred to in the text and shown in Figures 1 and 2. Both of these figures should be annotated so that the elements and changes being discussed at each stage are readily observable in the figures.

We realize that most readers will not be familiar with the 3D morphology and complexity of the chondrocranium and have provided new figure supplements to annotate the changes summarized in Figures1 and 2 for each age analyzed. Figures 1 and 2 and all figure supplements now label the cartilages to make our descriptions clear. Photographs of the cleared and stained specimens along with the 3D reconstructions of the chondrocrania in Figure 2 were intended to provide a more familiar representation of these structures and to make the anatomy accessible and understandable for readers who do not know the 3D morphology of the vertebrate chondrocranium and who are unfamiliar with ontogenetic changes of the chondrocrania in the two genotypes. We have now labeled structures in the cleared and stained specimens and in the segmented chondrocrania to clarify this anatomy, presenting it in a context more familiar to developmental biologists.

The changes in cellular parameters in the chondrocranium are not intuitively understood. For example, what explains the decrease in chondrocyte number at e15.5 in the mutants? It appears to greatly increase again at e16.5. There are other examples in this figure that are also confusing. Can these dynamic changes be explained biologically, or are they a result of the sampling?

We are confident that our sampling within defined regions of interest accurately represents the genotypes being compared as well as individual variation within these genotypes. Mechanisms controlling the activity of chondrocytes in the identified developing regions of interest are multifaceted and time sensitive. When, for example, the size of cells is significantly decreased in the lateral wall of *Fgfr2c^C342Y/+^* embryos relative to *Fgfr2c^+/+^* embryos at E13.5, there is a simultaneous significant increase in matrix area and chondrocyte number in *Fgfr2c^C342Y/+^* embryos. Most instances of a significant relative decrease in chondrocyte number, size, or matrix area in *Fgfr2c^C342Y/+^* embryos can be directly associated with significant increase in other measures of chondrocyte and cartilage size in *Fgfr2c^C342Y/+^* embryos at the same time. However, there are locations and instances when this does not occur (matrix area in braincase floor at E15.5), demonstrating that statistically insignificant changes in some cellular characteristics can have additive or interactive effects of biological significance, potentially offsetting the identified statistically significant changes in *Fgfr2c^C342/Y/+^* embryos. Though we agree with the reviewer that the changes we are identifying in chondrocyte number, especially within the braincase floor, may be the result of uneven distribution of chondrocytes at different stages of differentiation, such a specific determination requires a more nuanced assessment than what we have done here, and is outside of the scope of this study. A future manuscript investigating the activity (proliferation, differentiation, and cell cycle regulation) of cells across the dynamic developing braincase floor for example, is required to aid in our understanding of the complexity of how cells build morphology. However, ontogenetic change in the overall shape of the chondrocranium is complex, is the result of independent, integrated, and potentially additive effects of dynamic changes at the cellular level, and may involve cellular dynamics of which we are not currently aware. Our study includes a fairly standard histomorphometric analysis and discussion of the observed cellular changes by summarizing multiple factors currently known to be responsible for cartilage shape and size rather than focusing on individual significant differences within our histological assessment. We and others (https://www.nature.com/articles/s41598-021-87260-5) have found that FGF signaling has a tissue- and stage-specific role in craniofacial cartilage development, but we also acknowledge that other factors and pathways are vital for development. To aid the reader in appreciating these complexities, we have expanded our discussion of our quantitative results (lines 321-349) and added new explanatory sentences to this resubmission (lines 350-356).

Reviewer #3 (Recommendations for the authors):Bony fish, amphibians, lizards, birds, and other mammals should be included to address the 400 million years of cooperation between chondrocranium and dermatocranium.

You are correct. Due to the length limitations of the manuscript, your comments that the writing is generally too long and redundant, and the opinions of other reviewers, we have reoriented the focus of this paper away from evolution of the dermatocranium and chondrocranium and focus on the interaction of these two structures in embryonic development.